# Learning-Augmented Online Covering Problems

**Afrouz Jabal Ameli** [1]  **Laura Sanità** [2]  **Moritz Venzin** [2]

## Abstract

We give a very general and simple framework to incorporate predictions on requests for online covering problems in a rigorous and black-box manner. Our framework turns any online algorithm with competitive ratio $\rho(k, \cdot)$ depending on $k$, the number of arriving requests, into an algorithm with competitive ratio of $\rho(\eta, \cdot)$, where $\eta$ is the prediction error. With an accurate enough prediction, the resulting competitive ratio breaks through the corresponding worst-case online lower bounds, and smoothly degrades as the prediction error grows. This framework directly applies to a wide range of well-studied online covering problems such as facility location, Steiner problems, set cover, parking permit, etc., and yields improved and novel bounds.

## 1. Introduction

In recent years, there has been significant work to incorporate *ML-advice* in the design of *online algorithms*. The basic premise is to provide additional information to an online algorithm to improve its guarantees. Ideally, whenever the prediction is correct, we expect to break through worst-case lower bounds, and the guarantees should degrade smoothly with the prediction error. This paradigm has been applied (quite successfully) to a wide array of online problems, ranging from caching (34; 6), scheduling (32; 33), graph algorithms (44; 9) and matching/secretary problems (19), to name a few. The rapid evolving of this research direction, often referred to as *learning-augmented algorithms*, is evidenced by the collection of papers in the dedicated website (1). Given the attention that the field has gained in the past years, designing general approaches for learning-

augmented algorithms is of crucial interest for the community. In this paper, we contribute significantly to this goal by providing a new framework which applies to a general class of online *covering* problems.

Formally, a covering problem is given by some ground set $X'$, as well as a family of *covers* $\mathcal{S}$. Each $S \in \mathcal{S}$ indicates whether an element $e \in X'$ is covered by cover $S$ (hence, one can often identify a cover with a subset of $X'$). There is a cost function $\mathsf{cost} : 2^{\mathcal{S}} \to \mathbb{R}_{\geq 0}$ associated to the covers, and the goal is to find the *cheapest* sub-family $\mathcal{S}' \subseteq \mathcal{S}$ of covers to cover all ground set $X'$. We make the following natural assumptions (that justify the term *covering*): if $\mathcal{S}^1 \subseteq \mathcal{S}$ covers $X^1 \subseteq X'$ and $\mathcal{S}^2 \subseteq \mathcal{S}$ covers $X^2 \subseteq X'$, then (i) $\mathcal{S}^1 \cup S^2$ covers $X^1 \cup X^2$, and (ii) $\mathsf{cost}(\mathcal{S}' \cup \mathcal{S}'') \leq \mathsf{cost}(\mathcal{S}') + \mathsf{cost}(S'')$.

In the *online* setting considered in this paper, only a (unknown) subset of $X \subseteq X'$ requests needs to be actually covered. The elements of $X$ to be covered are revealed one-by-one, and we have to maintain a valid covering for the arrived elements in a monotone fashion (i.e. the decisions which sets to include are irrevocable). The goal is to minimize the *competitive ratio*. An online algorithm $\mathsf{ALG}$ for a *minimization* problem is $\rho$-competitive, if for any input $I$, the cost of the solution output by the online algorithm (denoted as $\mathsf{cost}(\mathsf{ALG}(I))$) is at most $\rho$ times the best solution for $I$ (denoted as $\mathsf{cost}(\mathsf{OPT}(I))$).

Many classical (online) optimization problems can be cast in the above form. Of course, the most natural one is the standard Set Cover problem. However, as we will discuss later, other examples include Facility Location problems, Steiner problems and Path Augmentation problems, among others. Most notably, these problems can be explicitly cast as instances of covering problems, though often with an exponential (in $|X|$) number of sets. Despite this, our method is powerful enough to remain valid. The main condition we need is simply the covering property defined above: if a partial solution is feasible for the currently revealed elements (that corresponds, e.g., to a subset of demands or connectivity requests when considering facility location or graph problems), it remains feasible after adding new sets (e.g., facilities or edges) to accommodate newly arrived elements.

Our framework applies to the following natural *learning-augmented* setting for online covering problems. We as-

[1]Department of Computer Science, Utrecht University, The Netherlands. [2]Department of Computing Sciences, Bocconi University, Milan, Italy. Correspondence to: Afrouz Jabal Ameli <a.jabalameli@uu.nl>, Laura Sanità <laura.sanita@unibocconi.it>, Moritz Venzin <moritz.venzin@gmail.com>.

*Proceedings of the $43^{rd}$ International Conference on Machine Learning*, Seoul, South Korea. PMLR 306, 2026. Copyright 2026 by the author(s).

sume that we receive upfront a prediction $\hat{X}$ on the elements that will eventually arrive. E.g. for the facility location problem, this prediction corresponds to arriving customers, or, for Steiner problems, to vertices that will eventually need to be connected to some root. This prediction is not necessarily completely accurate, and we define the *prediction error* $\eta$ as the symmetric difference between $\hat{X}$ and the set of actually arriving requests $X$, capped at $|X|$. Formally, $\eta := \min(|X|, |X \triangle \hat{X}|)$, where $X \triangle \hat{X} := (X \setminus \hat{X}) \cup (\hat{X} \setminus X)$. This is sometimes referred to as the $\ell_1$ error. We stress that our prediction error $\eta$ depends only on the *difference* between the set of actually arriving elements and the set of predicted elements. The actual ordering (in the arrival) is *not* important. In the following, we let $k := |X|$. Our framework shows that any online covering algorithm can be turned into a learning-augmented algorithm: provided there is an algorithm with competitive ratio depending on $k$, the number of actually arriving elements (e.g. $O(\log k \log |\mathcal{S}|)$ for set cover, $O(\log k)$ for Steiner tree), we obtain a learning-augmented online algorithm with a dependency on $\eta$, the prediction error ($O(\log \eta \log |\mathcal{S}|)$, $O(\log \eta)$ resp.). We emphasize that our approach operates in a fully black-box manner and does not rely on any particular feature of the online covering algorithm.

**Theorem 1.1** (Existence). *Let* ALG *be an algorithm for an online covering problem with a competitive ratio $\rho(k, \cdot)$, where $k$ is the number of arriving requests. Then, given a prediction $\hat{X}$, there exists an online algorithm with a competitive ratio of $O(\rho(\eta, \cdot))$.*

Our result demonstrates that for any online covering problem, it is always possible to incorporate predictions and obtain a refined competitive ratio that depends on the prediction error $\eta$. Furthermore, the resulting learning-augmented online algorithm is both *smooth* and *robust*: the competitive ratio $\rho(\eta, \cdot)$ improves over the baseline online algorithm whenever the prediction is (close to) perfect and smoothly degrades with the prediction error, but is never worse, even when the predictions are completely off (recall that $\eta \leq k$, by definition).

Our framework is based on a charging scheme that relies on a subtle *decomposition* of the solution space. Given some prediction $\hat{X}$, we compute *offline* partial solutions $\mathcal{S}_1, \mathcal{S}_2, \ldots \subseteq \mathcal{S}$, such that their union is feasible for $\hat{X}$. In the online phase, we use a $\rho(k, \cdot)$-competitive algorithm to satisfy the requests in a black-box fashion. We only keep track of its expenses: whenever its cost exceeds that of some partial solution $\mathcal{S}_i$, we buy $\mathcal{S}_i$ (in addition to whatever the online algorithm buys). Interestingly, this simple framework turns out to be powerful enough to guarantee a competitive ratio that depends on the prediction error, rather than the actual request sequence.

We now turn to discussing the *efficiency*. In fact, whenever the decomposition mentioned above can be computed efficiently, then the overall online algorithm turns into a polynomial time algorithm. However, while such a decomposition always exists (yielding the existential result), it is not always possible to compute it in polynomial time. This is not surprising, as there are covering problems for which an $O(\rho(\eta, \cdot))$-competitive algorithm *cannot* be achieved in polynomial-time, assuming $\mathsf{P} \neq \mathsf{NP}$. An example of this is the classical Set Cover problem (see Theorem 3.3). Nevertheless, we are still able to provide a *relaxed* notion of decomposition, that we call an $(\alpha, \gamma)$-*decomposition* (details in Section 2). This relaxed notion allows us to smoothly 'compromise' between the competitive ratio and the efficiency of the overall framework. The resulting guarantees are resumed in the following theorem.

**Theorem 1.2** (Efficiency). *Let* ALG *be an algorithm for an online covering problem with a competitive ratio $\rho(k, \cdot)$, where $k$ is the number of arriving requests. Assume that an $(\alpha, \gamma)$-decomposition for the* offline *version of the covering problem can be computed in polynomial-time. Then, given a prediction $\hat{X}$, there exists a polynomial-time online algorithm with a competitive ratio of*

$$O(\alpha) \cdot \rho(\eta, \cdot) + O(\alpha \cdot \log^{-1}(\tfrac{\gamma}{\gamma-1}) \cdot \log k).$$

*In particular, if $\gamma = 1$, the competitive ratio becomes $O(\alpha \cdot \rho(\eta, \cdot))$.*

Note that a $(1, 1)$-decomposition always exists and yields Theorem 1.1. This *additive* term bypasses the aforementioned impossibility to turn any $\rho(k, \cdot)$-competitive algorithm into a learning-augmented, $O(\rho(\eta, \cdot))$-competitive algorithm in polynomial time.

We demonstrate the versatility of our approach, by applying it to the vast class of online covering problems. In particular, our framework yields the *first* learning-augmented algorithms when considering prediction on requests, for classical problems like set cover, non metric facility location, path augmentation, and hard generalizations of Steiner problems.

We conclude this introduction by giving an account of example applications (Section 1.1), and a comparison to related work (Section 1.2). Our framework is formally presented in Section 2, and the details for our example applications are discussed in Section 3. In Section 4 we present the experimental evaluation of our approach for the online set cover problem.

All missing details and proofs can be found in the full version attached at the end.

## 1.1. Applications

**Set Cover** An instance $(X', \mathcal{S}, w)$ of online set cover is given by a ground set $X'$, an ensemble of sets $\mathcal{S} \subseteq 2^{X'}$, as well as a cost function $w : \mathcal{S} \to \mathbb{R}_{\geq 0}$. A subset of the elements $e \in X \subseteq X'$ arrive one-by-one. The goal is to maintain a subcollection $\mathcal{S}' \subseteq S$ covering all $X$, while minimizing $\sum_{S \in \mathcal{S}'} w(S)$. Instances of Set Cover naturally translate into instances $(X', \mathcal{S}, \mathsf{cost})$ of our covering problem by letting $\mathsf{cost}(\mathcal{S}') = \sum_{S \in \mathcal{S}'} w(S)$ for all $\mathcal{S}' \subseteq \mathcal{S}$. Set Cover admits a *randomized* online algorithm with competitive ratio $O(\log k \log |\mathcal{S}|)$ (4). Our framework yields the first learning-augmented algorithm for predictions on request, with a $O(\log \eta \log |\mathcal{S}|)$-competitive ratio, plus an additive $O(\log k) \cdot \mathsf{OPT}$, if it is required to run in polynomial time.

**Facility Location** An instance of (non-metric) Facility Location (NMFL) is given by a client-facility graph $(C', F)$, opening cost $w : F \to \mathbb{R}_{\geq 0}$, assignment cost $\mathsf{dist} : C' \times F \to \mathbb{R}_{\geq 0} \cup \{+\infty\})$. Clients $c \in C \subseteq C'$ are released one-by-one. The goal is to maintain a set $F'$ of open facilities and an assignment $x : C \times F' \to \{0, 1\}$ of arrived clients to open facilities, minimizing $\sum_{f \in F'} w(f) + \sum_{c \in C, f \in F'} \mathsf{dist}(c, f) x(c, f)$. Instances of Facility Location translate into instances $(X', \mathcal{S}, \mathsf{cost})$ of our covering problem by letting $X' = C'$, and by defining a set $S \in \mathcal{S}$ for every possible subset $F' \subseteq F$ and every possible assignment $\tilde{x}$ of a subset $\tilde{C}$ of clients to $F'$. The $\mathsf{cost}(S)$ of $S$ would be $\sum_{f \in F'} w(f) + \sum_{c \in \tilde{C}, f \in F'} \mathsf{dist}(c, f) \tilde{x}(c, f)$, and for any $\mathcal{S}' \subseteq \mathcal{S}$, $\mathsf{cost}(\mathcal{S}') = \sum_{S \in \mathcal{S}'} \mathsf{cost}(S)$. For this problem, there exists a *randomized* online algorithm with competitive ratio $O(\log k \log |F|)$ (4). Our framework yields a $O(\log \eta \log |F|)$- competitive algorithm, plus an additive $O(\log k) \cdot \mathsf{OPT}$, if it is required to run in polynomial time. To the best of our knowledge, our algorithm establishes the first learning-augmented algorithm under prediction on request for Facility Location. Previously, only the *metric* variant (that is, when the assignment function (dist) is a metric function) was studied (9). For this setting, the best competitive ratio is $O(\frac{\log k}{\log \log k})$, (35; 20). Our framework yields a polynomial-time, $O(\frac{\log \eta}{\log \log \eta})$-competitive learning-augmented algorithm. We elaborate in Section 1.2 how this compares with the result of (9), as their result holds with respect to a slightly different error measure, and most importantly, cannot be applied in a complete black-box fashion.

**Parking Permit and Weighted Path Augmentation** An instance of weighted path augmentation (WPAP) is given by $(E, \mathcal{L}, \mathsf{cost} : \mathcal{L} \to \mathbb{R}_{\geq 0})$. $E$ is a set of elements, that correspond to edges of a path $P$, indexed from 1 to $n$, and $\mathcal{L}$ is a set of *links*, weighted according to a cost function. Each link covers a consecutive interval of elements (subpath of $P$). In the online phase, elements arrive one-by-one, and we need to immediately and irrevocably select a link covering

it. This can be thought as a connectivity augmentation problem where edges of a path can fail online, and we need to restore connectivity. Instances of this problem map into instances $(X', \mathcal{S}, \mathsf{cost})$ of our covering problem by letting $X' = E, \mathcal{S} = \mathcal{L}$. The Parking Permit problem is a special case of WPAP. For any element (edge of the path), we can select $K$ different permits (links) of fixed durations (i.e. length). For both problems, it is possible to devise a deterministic algorithm that is $O(\log k)$-competitive, as well as a randomized online algorithm that is $O(\log \log k)$-competitive against oblivious adversaries. Our results yield a deterministic, $O(\log \eta)$-competitive online algorithm and a randomized, $O(\log \log \eta)$-competitive online algorithm. Again, these are the first learning-augmented algorithms for connectivity augmentation problems. We provide the details in Section 3.2.

**Steiner Tree and variants** In Steiner tree (resp. forest) problems, given a metric graph $G$, terminals $\{t_1, t_2, \ldots, t_k\}$ show up one-by-one (resp. in pairs), and one needs to maintain a connected subgraph connecting all arrived terminals (terminal pairs). Instances of this problem map into instances $(X', \mathcal{S}, \mathsf{cost})$ of our covering problem by letting $X' = V$, $\mathcal{S}$ be the set of all possible $t_i - t_j$ paths in the graph, and by defining $\mathsf{cost}(\mathcal{S}')$ to be the sum of edge (or nodes) weights in the union of the corresponding paths. In the edge-weighted setting, there are $O(\log \eta)$-competitive algorithms for these two problems (44; 9), which we can also obtain with our framework. However, for several important variations such as group Steiner tree / forest, as well as the node-weighted and directed setting, no learning-augmented algorithms were known prior to our work. For these problems, the competitive ratio is of order $O(\log^{O(1)} k \log^{O(1)} |V|)$, (37; 16). Our framework directly improves this to $O(\log^{O(1)} \eta \log^{O(1)} |V|)$ in the learning-augmented setting. We provide the details in Section 3.3.

## 1.2. Comparison to related work and techniques

The study of general schemes in the context of learning-augmented online algorithms for set cover (and other covering problems) was initiated by (11). They devise a primal-dual approach for covering problems that incorporates predictions. Subsequent works on the problem then consider further variations, see for instance (5; 24; 10). However, a common feature of these works, is that the prediction is on the sets, i.e. the online algorithms receives hints on what an optimal solution (conditioned on the set of actually arriving elements) looks like. As such, these results are incomparable to ours. Despite this, we would like to mention that the results in (11; 23; 10) could still be *combined* with our framework (see full version for more details).

A general framework that is close to ours is the one by (9). Here they study an interesting generalization of the error we

consider, specialized to problems on metric graphs. Specifically, they consider the *metric error with outliers* given by $(\triangle, D)$: for some $\hat{T} \subseteq \hat{X}$, $T \subseteq X$ with $|T| = |\hat{T}|$, and a min-cost matching between $T$ and $\hat{T}$ of cost $D$, and $|\hat{X} \setminus \hat{T} \cup X \setminus T| = \triangle$. An important feature of their work, is that their guarantees hold with respect to the best choice of $\triangle$ and $D$. In particular, setting $\hat{T} = T = \hat{X} \cap X$ (consequently $D = 0$), their error measure is always smaller than $\eta$ for metric graph problems. On a high-level, the overall approach is similar as they also compute partial solutions to the previously arrived elements, and pay towards predicted requests. This is decided based on an instance of a *prize-collecting* variant of the problem. However, to obtain their guarantees, they require the underlying online algorithm to be *subset-competitive*, a non-trivial property (for instance for some online algorithms for the metric facility location problem). In contrast, we rely on a *more involved* decomposition, but our results *do not* depend on specifics of the underlying online algorithm and can be applied in a completely modular and black-box way. As a direct consequence, we can also apply our results to any standard algorithm for these problems. In particular, by leveraging the *best* known online algorithm for metric facility location with competitive ratio $O(\frac{\log k}{\log \log k})$, we obtain a bound of $O(\frac{\log \eta}{\log \log \eta})$. In comparison, they give a $O(\log \triangle)$-competitive ratio (plus an additive term of $D$), as they rely on a $O(\log k)$-competitive algorithm to ensure subset-competitiveness. As a side note, we remark that there has been significant work on learning-augmented online facility location, see (5; 3; 29; 10; 21). However, all these works assume that predictions are hints on the location of optimal facilities. As such, these results are incomparable to ours. The framework in (9) also applies to Steiner tree and Steiner forest, yielding a $O(\log \triangle)$-(plus additive $D$) competitive ratio (a bound of $O(\log \eta)$ for Steiner tree was previously given in (44)). Our framework recovers this latter guarantee of $O(\log \eta)$, but is also applicable for the generalisations in Section 1.1.

For the parking permit and its generalization, the weighted path augmentation problem, there were no known learning-augmented algorithms. The parking permit was introduced in (36), where a deterministic $O(\log n)$-, and a randomized, $O(\log \log n)$-competitive algorithm was given. However, it is straightforward to adapt this algorithm to yield a $O(\log k)$- (resp. $O(\log \log k)$-) competitive algorithm for parking permit. A generalisation of WPAP was first studied by (38), and they provide a deterministic, $O(\log n)$ competitive algorithm for the problem, as well as a fractional, $O(\log \log n)$-competitive algorithm for WPAP. We adapt these ideas to yield a deterministic, $O(\log k)$-competitive algorithm for WPAP, as well as a randomized, $O(\log \log k)$-competitive algorithm, for which our framework can be applied.

We conclude this subsection with a final remark. The error $\eta$ we consider is essentially the $\ell_1$ error, the term $\min\{|X|, \cdot\}$

simply follows from robustification. As such, our error measure is very simple and efficient to learn, it suffices to take the element-wise majority over a *logarithmic* number of past instances (see full version). It also naturally handles fractional predictions, where each request is associated with some probability or confidence parameter, as these can be rounded while preserving the overall error.

## 2. Framework for predictions on requests

We now describe ICE (**I**teratively **C**harge **E**xpenses), a deterministic online procedure to incorporate predictions on requests. This procedure operates in two stages. In the first, offline phase, we receive the prediction $\hat{X}$ on the requests. We compute a decomposition for $\hat{X}$, i.e.

$$\hat{X}_1 \sqcup \hat{X}_2 \sqcup \ldots \sqcup \hat{X}_n = \hat{X},$$

and corresponding feasible solutions $\mathcal{S}_1, \mathcal{S}_2, \ldots, \mathcal{S}_n$, of respective costs $c(\mathcal{S}_i)$. In the online phase, we keep track of the cost by ALG incurred on elements in $X \cap \hat{X}$. Whenever this exceeds the cost of $\mathcal{S}_i$, we buy $\mathcal{S}_i$. We resume this in Algorithm 2.

---

**Algorithm 1** ICE - Iteratively Charge Expenses

---

1: **Input:** Online algorithm ALG with competitive ratio $\rho(k, \cdot)$ and predicted requests $\hat{X}$.
2: **Offline phase:** Compute $(X_i, \mathcal{S}_i)$s as detailed in Section 2.1.
3: Initialize two instances of ALG, $\mathsf{ALG}_+$ and $\mathsf{ALG}_-$.
4: Set excess $\leftarrow 0$, layer $\leftarrow 1$.
5: **when** request $x \in X$ arrives
6:     **if** $x \notin \hat{X}$ **then**
7:         Pass $x$ to $\mathsf{ALG}_+$.
8:     **else**
9:         Pass $x$ to $\mathsf{ALG}_-$. Increase excess by the cost of the partial solution $\mathsf{ALG}_-$ selects for $x$.
10:         **while** excess $\geq c(\mathcal{S}_{\mathsf{layer}})$ **do**
11:             Buy $S_{\mathsf{layer}}$, layer $\leftarrow$ layer $+ 1$, excess $\leftarrow$ excess $- c(\mathcal{S}_{\mathsf{layer}})$ and reinitialize $\mathsf{ALG}_-$.
12:         **end while**
13:     **end if**
14: **end when**

---

The running time of our procedure is at most that of the online algorithm that it uses in a black-box manner, as well as the time to compute the decomposition in Line 2. We formalize this in the following subsection.

### 2.1. Offline phase of the General Framework

We first illustrate our approach. To this end, assume that we know the exact value of $\eta$ and that $X \subseteq \hat{X}$.[1] Our

---

[1] There are at most $\eta$ elements in $X \setminus \hat{X}$, so we can use a separate online algorithm for these elements that is $\rho(\eta, \cdot)$-competitive.

framework can be simplified to the following two steps: in the preliminary (offline) stage, we compute $\tilde{S}$, a minimum cost collection of sets, that covers $|X| - \eta$ elements of $X$. In the online part of the algorithm, we use the $\rho(k, \cdot)$-competitive algorithm in a black-box fashion. Once the cost of selected sets exceeds $\tilde{S}$, we buy all sets belonging to $\mathcal{S}$. To see why this results in an $O(\rho(\eta, \cdot))$-competitive algorithm, observe that the cardinality of the elements in $X$ that are not covered by a set in $\tilde{S}$, is at most $\eta$. Hence on these $\eta$ elements, the online algorithm is $O(\rho(\eta, \cdot))$-competitive. On the other hand, on the elements covered by $\tilde{S}$, the online algorithm can never spend more than OPT: this is since the cost of $\tilde{S}$ is a lower bound on the optimum, and we buy all sets belonging to $\tilde{S}$ once the online algorithm spends more than the cost of $\tilde{S}$.

Unfortunately, it seems there is no direct way to obtain an estimate on $\eta$, standard techniques like guess-and-double do not work. Instead, we take a different route. We construct sets $\mathcal{S}_i$, whose union progressively gets 'closer' to the (unknown) set $\tilde{S}$. Indeed, it is not necessary to exactly find $\tilde{S}$, but is sufficient to approximate it in terms of cardinality and total cost. To handle both, we impose some geometric growth condition on the sets $\mathcal{S}_i$, see properties (A) and (B) below. Each $\mathcal{S}_i$, analogously to $\tilde{S}$, is the cheapest collection of sets covering some prescribed fraction of (remaining) elements. Computing these sets is typically an NP-hard problem, which requires approximation to remain tractable. To this end, we introduce the notion of $(\alpha, \gamma)$-decomposability, see Definition 2.1, which we use in Properties (C) and (D) of the decomposition to control the total cost of these sets $\mathcal{S}_i$, and bound the resulting competitive ratio.

**Definition 2.1** (Efficient $(\alpha, \gamma)$-decomposable)**.** An offline covering problem with request set $X$ is said to be $(\alpha, \gamma)$-decomposable if for all $i \in \{1, 2, \ldots, |X|\}$, we can compute a set $B \subseteq X$ and a collection $\mathcal{S}_B$ covering $B$, s.t.

- The cost of $\mathcal{S}_B$ is at most $\alpha$ times the minimum cost of covering $i$ requests in $X$, i.e.
$$c(\mathcal{S}_B) \leq \alpha \cdot \min_{A \subseteq X, |A|=i} \{c(\mathcal{S}_A) \mid \mathcal{S}_A \text{ covers } A\}, \text{ and,}$$

- $|B| \geq i/\gamma$.

Whenever $B$ and $\mathcal{S}_B$ can be computed in polynomial time, we say the problem is *efficiently* $(\alpha, \gamma)$-*decomposable*.

Note that any problem is $(1, 1)$-decomposable, e.g. through integer programming or enumeration techniques. However, this may take super-polynomial time, as is the case, e.g., for set cover.

Next, we use this property of $(\alpha, \gamma)$-decomposability, to obtain an $(\alpha, \gamma)$-*decomposition*. For convenience, we set $g_{\alpha,\gamma}(t) := \alpha$ if $\gamma = 1$ and $g_{\alpha,\gamma}(t) := \alpha \cdot (1 + \log^{-1}(\frac{\gamma}{\gamma-1}) \cdot \log t)$ if $\gamma > 1$.

We split $\hat{X}$ through an iterative process: starting with $R_0 := \hat{X}$, we select a subset $X_1 \subseteq R_0$, update $R_1 \leftarrow R_0 \setminus X_1$, and recurse with $R_1$. Then for every $i$, we denote by $\mathcal{S}_i$ the ($\alpha$-approximate) solution computed to cover the elements in $X_i$, as in Definition 2.1. An $(\alpha, \gamma)$-decomposition is then a pair $(X_i, \mathcal{S}_i)$, where $\hat{X} = X_1 \cup X_2 \cup \cdots \cup X_r$, and the following properties hold for each $i$:

(A) $|X_i| \geq \frac{|\hat{X}| - (|X_1| + \ldots + |X_{i-1}|)}{2}$.

(B) If $c(\mathcal{S}_i) < 2c(\mathcal{S}_{i-1})$, then $8c(\mathcal{S}_{i-1}) < c(\mathcal{S}_{i+1})$.

(C) If $c(\mathcal{S}_i) > 10c(\mathcal{S}_{i-1})$, then $c(\mathcal{S}_i) \leq g_{\alpha,\gamma}(|\hat{X}|) \cdot C_1$, where $C_1$ is the minimum cost solution that covers $\lceil \frac{|\hat{X}| - (|X_1| + \ldots + |X_{i-1}|)}{2} \rceil$ elements from $R_{i-1}$.

(D) $c(\mathcal{S}_i) \leq g_{\alpha,\gamma}(|\hat{X}|) \cdot C_2$, where $C_2$ is defined as follows: for $i \geq 2$, $C_2$ is the minimum cost solution that covers $|X_i|$ elements from $R_{i-1}$; for $i = 1$, $C_2$ is the minimum cost solution that covers $\lceil |\hat{X}|/2 \rceil$ elements from $R_0$.

**Theorem 2.2.** *There is a polynomial-time construction guaranteeing properties (A), (B), (C), and (D), provided that the problem is efficiently $(\alpha, \gamma)$-decomposable.*

The proof is in the full version.

## 2.2. General Framework Analysis

In this section, we analyze the competitive ratio of Algorithm 2. Assume that upon the termination of our algorithm (Algorithm 2), we are at layer $i + 1$ and $\overline{\text{excess}}$ is the value of excess when the algorithm ends. That is, in the previous steps we were able to buy $\mathcal{S}_1 \cup \ldots \cup \mathcal{S}_i$, and layer $= i + 1$ when the algorithm terminates, with $\overline{\text{excess}} < c(\mathcal{S}_{i+1})$. We also define $\Delta_- := |\hat{X} \setminus X|$.

The following two lemmata give a lower bound on the size of the optimal solution.

**Lemma 2.3.** *Let $m$ be a positive integer. If $\Delta_- \leq |R_m|$, then OPT $\geq \max_{1 \leq j \leq m} \{c(\mathcal{S}_j)\}/g_{\alpha,\gamma}(|\hat{X}|)$.*

*Proof.* Let $j \in \{1, \ldots, m\}$. As $\Delta_- \leq |R_m|$, then $\Delta_- \leq |R_j|$. This means that at least $|X_j|$ many elements from $R_{j-1}$ appear in our instance (otherwise $\Delta_- > |R_j|$ as $|R_j| + |X_j| = |R_{j-1}|$). So, by Property (D), the cost to cover these elements is at least $c(\mathcal{S}_j)/g_{\alpha,\gamma}(|\hat{X}|)$. $\square$

**Lemma 2.4.** *The following holds:*

- *For every $j \in \{1, \ldots, i\}$, OPT $\in \Omega(c(\mathcal{S}_j)/\rho(\Delta_-, .))$ or OPT $\in \Omega(c(\mathcal{S}_j)/g_{\alpha,\gamma}(|\hat{X}|))$, and*

- OPT $\in \Omega(\overline{\text{excess}}/\rho(\Delta_-, \cdot)$ *or* OPT $\in \Omega(\overline{\text{excess}}/g_{\alpha,\gamma}(|\hat{X}|))$.

*Proof.* For every $j \in \{1, \ldots, i+1\}$, let us denote by $a_j$ the minimum of the value of excess at the beginning of layer $j$ and the value of $c(\mathcal{S}_j)$. Also for every $j \in \{1, \ldots, i\}$ we define $c(\mathsf{ALG}_j) := c(\mathcal{S}_j) - a_j$. For $j = i + 1$, we define $c(\mathsf{ALG}_j) := \overline{\text{excess}} - a_j$. Consequently, we have $a_1 = 0$, and $0 \leq a_j \leq c(\mathcal{S}_j)$.

To better explain the above notation, consider the following scenario; Assume we are at layer $= j$, and then an element $x \in \hat{X}$ arrives. According to line 9 of Algorithm 2, ALG buys the partial solution $S$ for $x$. Now we update excess $\leftarrow$ excess $+ c(S)$. Now if excess $\geq c(\mathcal{S}_{\text{layer}})$, let $z$ be the largest number such that excess $\geq \sum_{i=\text{layer}}^{z} c(\mathcal{S}_i)$. In this case, we will have $a_{j+1} = c(\mathcal{S}_{j+1})$, $a_{j+2} = c(\mathcal{S}_{j+2})$, $\ldots$, $a_z = c(\mathcal{S}_z)$, and $c(\mathsf{ALG}_{j+1}) = c(\mathsf{ALG}_{j+2}) = \cdots = c(\mathsf{ALG}_z) = 0$; Also we will have $a_{z+1} = \text{excess} - \sum_{i=\text{layer}}^{z} c(\mathcal{S}_i)$.

To prove the claim for any $j$, we will show that OPT $\in \Omega(\min\{\frac{c(\mathsf{ALG}_j)}{\rho(\Delta_-, \cdot)}, \frac{c(\mathsf{ALG}_j)}{g_{\alpha,\gamma}(|\hat{X}|)}\} + a_j)$.

We first show that OPT $\geq a_j$. As $a_1 = 0$, the claim is trivial for $j = 1$. If $j \geq 2$ the cost that the online algorithm spends at the stage which led to purchasing $\mathcal{S}_{j-1}$ is at least $a_j$, and therefore $a_j$ is a lower bound on OPT; This is true as without loss of generality, ALG never pays more than OPT for any element. To complete the argument, we will show OPT $\in \Omega(\frac{c(\mathsf{ALG}_j)}{\max\{\rho(\Delta_-, \cdot), g_{\alpha,\gamma}(|\hat{X}|)\}})$. Note that the number of elements handled by our algorithm in layer $j$ is at most $|X_j|$, as otherwise by Property (D), the cost to cover only these elements is at least OPT $\geq c(\mathsf{ALG}_j)/g_{\alpha,\gamma}(|\hat{X}|)$.

So by the assumption on the competitive ratio of ALG we have OPT $\in \Omega(\frac{c(\mathsf{ALG}_j)}{\rho(|X_j|, \cdot)})$ (the online algorithm for elements that arrive at layer $j$, is an instance with at most $|X_j|$ elements and hence is $\rho(|X_j|, \cdot)$-competitive). This means that if $\Delta_- \geq |X_j|/2$ we are done as $\rho(\Delta_-, \cdot) \in \Omega(\rho(|X_j|, \cdot))$ (note that the competitive ratio can grow at most linearly). So we assume that $\Delta_- < |X_j|/2$. As $X_j \subseteq R_{j-1}$, then $\Delta_- \leq |R_{j-1}|$ and by Lemma 2.3, if $j > 1$ then we have OPT $\geq c(\mathcal{S}_{j-1})/g_{\alpha,\gamma}(|\hat{X}|)$. We distinguish two cases; If $j > 1$ and $c(\mathcal{S}_j) \leq 10c(\mathcal{S}_{j-1})$, then OPT $\geq c(\mathcal{S}_{j-1})/g_{\alpha,\gamma}(|\hat{X}|)$ already implies OPT $\in \Omega(c(\mathsf{ALG}_j)/g_{\alpha,\gamma}(|\hat{X}|))$, and hence the claim. Now consider the case $j = 1$ or $c(\mathcal{S}_j) > 10c(\mathcal{S}_{j-1})$; As $\Delta_- < |X_j|/2$, then we have $\Delta_- < |R_{j-1}|/2$. By Property (C) for $j > 1$ and Property (D) for $j = 1$, it holds that OPT $\geq c(\mathcal{S}_j)/g_{\alpha,\gamma}(|\hat{X}|)$. $\qquad\square$

In the next claim, we give an upper-bound on the cost of $\mathsf{ALG}_-$.

**Claim 2.1.** *The cost of $\mathsf{ALG}_-$ can be upper bounded by $O(c(\mathcal{S}_{i-1}) + c(\mathcal{S}_i)) + \text{excess}$.*

*Proof.* The cost of $\mathsf{ALG}_-$ is $O(\sum_{j=1}^{i} c(\mathcal{S}_j)) + \text{excess}$. By

Property (B), for every $j$, $c(\mathcal{S}_j) + c(\mathcal{S}_{j+1}) \leq \frac{3}{4}(c(\mathcal{S}_{j+2}) + c(\mathcal{S}_{j+3}))$. So $\sum_{j=1}^{i-2} c(\mathcal{S}_j) \in O(c(\mathcal{S}_{i-1}) + c(\mathcal{S}_i))$ and hence the claim. $\qquad\square$

Before proving our main result of this section, namely Theorem 2.6, we introduce some useful notation. We define $k_- : |X \cap \hat{X}|$ and $k_+ := |X \setminus \hat{X}|$ as the number of arriving elements that are handled by $\mathsf{ALG}_-$ and $\mathsf{ALG}_+$, respectively ($k_- + k_+ = k$). We first analyze the competitive ratio of $\mathsf{ALG}_-$.

**Lemma 2.5.** *Our algorithm for $\mathsf{ALG}_-$ is $O(\rho(\eta, \cdot) + g_{\alpha,\gamma}(k_-))$-competitive.*

*Proof.* We first assume $k_- \geq |\frac{\hat{X}}{2}|$. From Lemma 2.4 and Claim 2.1, we observe that our algorithm is $O(\rho(\Delta_-, \cdot) + g_{\alpha,\gamma}(|\hat{X}|))$-competitive. As $k_- \geq \frac{|\hat{X}|}{2}$, and by definition of $\Delta_-$, $\eta \geq \min\{k_-, \Delta_-\}$, and therefore $\rho(\eta, \cdot) + g_{\alpha,\gamma}(|\hat{X}|) \in O(\rho(\eta, \cdot) + g_{\alpha,\gamma}(k_-))$.

Now assume $k_- < |\hat{X}|/2$. In this case $k_- < \eta$, and it suffices to show that our algorithm is $O(\rho(k_-, .))$-competitive. For this purpose, we observe that at each layer $j$ of $\mathsf{ALG}_-$ our cost is at most $O(\rho(k_-, .))$OPT, as at most $k_-$ elements arrive. By Claim 2.1 our algorithm is $O(\rho(k_-, .))$-competitive. $\qquad\square$

Now we have all the ingredients to analyze the competitive ratio of Algorithm 2.

**Theorem 2.6.** *Algorithm 2 is $O(\rho(\eta, \cdot) + g_{\alpha,\gamma}(k))$-competitive.*

*Proof.* $\mathsf{ALG}_+$ is $O(\rho(k_+, \cdot))$-competitive as it is an instance of ALG with $k_+$ arriving elements. By Lemma 2.5, $\mathsf{ALG}_-$ is $O(\rho(\eta, \cdot) + g_{\alpha,\gamma}(k_-))$-competitive. By definition of $k_+$, it holds that $k_+ \leq \eta$ and hence $\rho(k_+, \cdot) \in O(\rho(\eta, \cdot))$. Therefore, Algorithm 2 is $O(\rho(\min\{\eta, k\}, \cdot) + g_{\alpha,\gamma}(k))$-competitive. $\qquad\square$

Theorems 1.1 and 1.2 follow from Theorems 2.6 and 2.2.

## 3. Applying our framework

### 3.1. Set Cover and Non-Metric Facility Location

The main theorem of this section is the following.

**Theorem 3.1.** *There is a polynomial-time, $O(\log \eta \log |\mathcal{S}| + \log k)$- (resp. $O(\log \eta \log |F| + \log k)$-) competitive algorithm for online set cover (resp. non-metric facility location). In exponential time, the dependence on $\log k$ can be removed.*

By Theorem 2.2 combined with Theorem 1.2, it is sufficient to show that these problems are efficiently $(\alpha, \gamma)$-decomposable. This is shown in Lemma 3.2. For the

exponential-time variant, we use Theorem 1.1 (observe once again that any covering problem is $(1, 1)$-decomposable).

**Lemma 3.2.** *Set cover and Non-metric Facility Location are efficiently $(1, e/(e − 1))$-decomposable.*

*Proof of Lemma 3.2.* By binary search, we may assume we know a number $C$, such that the optimal cost of covering $i$ elements of $(X, \mathcal{S})$ equals $C$. Convert $(X, \mathcal{S})$ into an instance of so-called *Budgeted Maximum Coverage Problem (BMC)*: In the BMC problem, given a collection $\mathcal{S}$ of sets defined over a universe of weighted elements, where each set $S \in \mathcal{S}$ has an associated cost $c(S)$, and given a budget $C$, the objective is to select a subcollection $\mathcal{S}' \subseteq \mathcal{S}$ such that the total cost satisfies $\sum_{S \in \mathcal{S}'} c(S) \leq C$, and the total weight of the elements covered by $\mathcal{S}'$ is maximized. Our set cover instance $(X, \mathcal{S})$ converts into a BMC instance with budget $C$, where each element has unit weight. Khuller et al. (30) presented a $(1 − 1/e)$-approximation algorithm for BMC, hence applying it we can cover at least $i \cdot \frac{e−1}{e}$ elements while not exceeding the budget $C$.

By the results from (26; 30; 43), similar arguments can be applied for NMFL (see full version). □

Note that this is asymptotically optimal, a $(1, e/(e−1)−\epsilon)$-decomposition in polynomial time implies a polynomial-time $(1 − O(\epsilon)) \ln |X|$-approximation to set cover, contradicting P $\neq$ NP, (18). Extending this, we show a corresponding lower bound for online set cover with predictions.

**Theorem 3.3.** *No polynomial-time learning-augmented online algorithm for set cover achieves an $O(\log \eta \log |\mathcal{S}|)$-competitive ratio (or $o(\log k)$).*

*Proof.* Consider an offline instance of set cover $(X, \mathcal{S})$. We use such a learning-augmented online algorithm to solve it (present the elements of $X$ in arbitrary order). This results in an approximation guarantee of $O(\log |\mathcal{S}|)$. Since set cover does not admit a $O(\log |\mathcal{S}|)$- or a $(1 − \epsilon) \cdot \ln |X|$-approximation in polynomial time (assuming P $\neq$ NP), see (39; 18), the proof follows. □

### 3.2. Weighted Path Augmentation and Parking Permit Problem

The main result of this section is the following.

**Theorem 3.4.** *There is a deterministic, $O(\log \eta)$- and a randomized, $O(\log \log \eta)$-competitive learning-augmented online algorithm for weighted path augmentation and parking permit.*

The laminar structure of WPAP allows us to exactly compute the decomposition.

**Lemma 3.5.** WPAP *is efficiently $(1, 1)$-decomposable.*

*Proof.* Consider an instance of WPAP with elements $E$, where the elements are indexed from $1$ to $n$. For every $1 \leq i \leq n$ and in poly($n$)-time, we show how to compute a subset $A \subseteq E$ of size at least $i$, and a set of links in $L \subseteq \mathcal{L}$ that covers $A$, such that cost($L$) is minimized. We handle this by dynamic programming. We define $\mathsf{DP}[i, j]$ to be the minimum cost to cover $i$ items from $j$ to $n$, only allowing links that do not contain elements from $1$ to $j − 1$. If this is infeasible, set the value to $+\infty$. Clearly, when $i = n−j+1$, this can be solved exactly, since the constraint matrix is totally unimodular, since it has the consecutive ones property (Chapter 19 of (40)). To update the table for other values, we proceed as follows. Denote by $c_{j,k}$ the cheapest cost of covering all elements between $j$ and $k$ (including the endpoints) with sets not including any element after $k$ (if $k < j$, this is 0). Again, $c_{j,k}$ can be computed exactly by total unimodularity. We set

$$\mathsf{DP}[i + 1, j] := \min_{j \leq k \leq j+i} c_{j,k−1} + \mathsf{DP}[i − (k − j), k + 1].$$

Correctness follows from the observation that if the $k^{th}$ element ($i \leq k \leq n$) is the first element not to be selected for a potential solution in $\mathsf{DP}[i + 1, j]$, the cost of covering elements $i$ to $k − 1$ is exactly $c_{j,k−1}$, and, the cost of selecting the remaining $i − (k − j)$ elements equals $\mathsf{DP}[i − (k − j), k + 1]$, since any link starting earlier than $k + 1$ includes $k$ and can be disregarded. The running time of this algorithm is poly($n$) as $i$, $j$ and $k$ are at most $n$. □

*Sketch of proof of Theorem 3.4.* We use Theorem 1.2 with $\alpha, \gamma = 1$, since by Lemma 3.5, WPAP is $(1, 1)$-decomposable. By the results in this paper (see full version), we provide a deterministic, $O(\log k)$-competitive algorithm, and a randomized, $O(\log \log k)$-competitive algorithm for WPAP. Consequently, our framework yields a learning-augmented deterministic (resp. randomized) $O(\log k)$-competitive (resp. $O(\log \log k)$) online algorithm. □

### 3.3. Metric facility Location and Steiner variants

The first result from this section is the following.

**Theorem 3.6.** *There is a polynomial-time, $O(\frac{\log \eta}{\log \log \eta})$-competitive learning-augmented algorithm for metric facility location.*

*Proof.* By the results of (35; 20), there is a deterministic $O(\frac{\log k}{\log \log k})$-competitive algorithm for facility location. Hence, this result follows in a black-box way from our framework, provided the problem is efficiently $(O(1), 1)$-decomposable. Decomposability follows from a problem that has been studied in the setting of *facility location with outliers*. In particular, in Theorem 5.2 of (17), the authors give a 3-approximation to the problem of serving a prescribed fraction of the clients as cheaply as possible,

meaning that a $O(3,1)$-decomposition can be computed in polynomial-time. □

We now list other prominent problems that have received considerable attention in the network design community, where our framework applies. With the exception of (edge-weighted) Steiner tree, we do not know how to turn our approach polynomial, i.e. how to compute an $(O(1),1)$-decomposition in polynomial time. Note that most of these problems generalize the set cover problem, hence this is impossible in polynomial-time.

**Theorem 3.7.** *Our framework provides a learning-augmented algorithm with competitive ratio $O(\rho(\eta,\cdot))$ for (i) Steiner tree and Steiner forest; (ii) Connected facility location; (iii) Group Steiner Tree and Forest; (iv) Node-weighted Steiner problems; (v) Directed Steiner problems.*

We instantiate our framework with the best known online algorithms for the corresponding underlying problems.

- Steiner tree and Steiner forest: These problems have $O(\log k)$-competitive online algorithms, (27; 12). These are the only two Steiner variants considered previously in the literature in our learning-augmented setting. Results were discussed in the introduction ((44; 9)).

- Connected facility location: This is a variation of the facility location problem, where the facilities need to remain connected. As such, it generalises the online Steiner tree problem. The best competitive ratio is $O(\log k)$, (42).

- Group Steiner Tree and Forest: The group setting is an important generalization of Steiner problems. Groups of terminals (terminal pairs) arrive one-by-one, and for each group, one needs to select one terminal (pair) to be included in the Steiner Tree (forest). For edge-weighted graphs, the best competitive ratio is of order $O(\log^7 k \log^5 |V|)$, (37).

- Node-weighted Steiner problems: This generalizes (edge-weighted) Steiner problems and set cover problems. Problems are defined analogously, but the costs are on vertices (and edges). To use an edge, one needs to also have purchased its two incident vertices. The best achievable competitive ratio for Steiner problems is typically a poly-logarithmic factor higher than the edge-weighted setting, (37; 14), and for generalizations such as the group-setting, the best-known running time is *quasi-polynomial*, i.e. of the form $|V|^{\mathsf{poly}(\log|V|)}$.

- Directed Steiner problems: This further generalizes the node- and edge-weighted setting. In polynomial-time, there is a $O(k^\epsilon \log^{O(1)}(|V|))$- $(O(k^{1/2+\epsilon} \log^{O(1)}(|V|))$-) competitive algorithm for directed Steiner tree (forest), for any $\epsilon > 0$, (16).

We remark that ours are the first learning-augmented algorithms for all these problems, with the exception of the (edge-weighted) Steiner tree/forest in (i) (for which we match the best known bound).

In the remainder of the section, we show how to recover the results from (44) (and partially (9)).

**Theorem 3.8.** *Our framework provides a polynomial-time learning-augmented $O(\log \eta)$-competitive online algorithm for online Steiner tree.*

We make use of the $k$-MST and $k$-Steiner tree problem.

**Definition 3.9** ($k$-MST Problem)**.** In the $k$-Minimum Spanning Tree problem, we are given an edge-weighted graph $G = (V, E)$, and a designated root vertex $r \in V$. The objective is to compute a tree rooted at $r$ that spans at least $k$ vertices while minimizing the total cost.

There exist several constant-factor approximation algorithms for the $k$-MST problem (see (7; 8; 13)), culminating in a 2-approximation algorithm presented in (22).

**Definition 3.10** ($k$-Steiner Tree Problem)**.** In the $k$-Steiner Tree problem, we are given an edge-weighted graph $G = (V, E)$, where the vertex set is partitioned into terminals $\{r\} \cup T$ and Steiner nodes $S$, i.e., $V = T \cup S$. The objective is to compute a tree rooted at $r$ that spans at least $k$ terminals while minimizing the total cost of its edges.

**Lemma 3.11.** *There exists a $4$-approximation for the minimum $k$-Steiner tree problem.*

*Proof.* We shortcut all Steiner nodes $S$, by passing to the *metric closure* restricted to terminals in $\{r\} \cup T$. That is, we consider the complete graph on vertex set $\{r\} \cup T$, where the cost of an edge between any pair of vertices corresponds to the cost of their shortest path in the original instance. Any tree can be mapped to the original instance with no greater cost. Conversely, by passing to the metric closure, the cost of any tree on terminals is increased by at most a factor of 2, (31). Hence, using the 2-approximation for $k$-MST with root $r$ on the metric closure, and mapping back to the original graph, connects at least $k$ terminals with the root, and has total cost at most 4 times that of a minimum $k$-Steiner tree. □

*Proof of Theorem 3.8.* Lemma 3.11 shows that this problem is efficiently $(4,1)$-decomposable. As there exists a $O(\log k)$-approximation for the Online Steiner Tree Problem (28), by Theorem 1.2 our framework provides a $O(\log \eta)$-competitive online algorithm for Steiner tree. □

# 4. Experimental evaluation

In this section, we evaluate the performance of our framework for the set cover problem. Our choice of set cover for experimental evaluation is motivated by practical considerations. The decomposition algorithms are efficient and admit simple implementations. Additionally, the existence of an accessible dataset makes Set Cover well suited for conducting scalable and meaningful experiments.

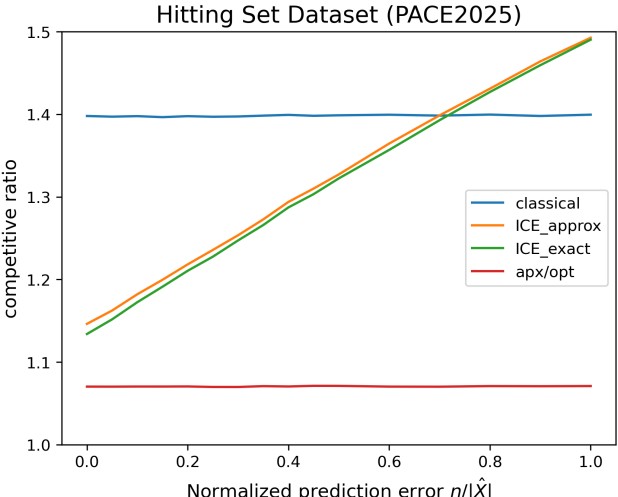

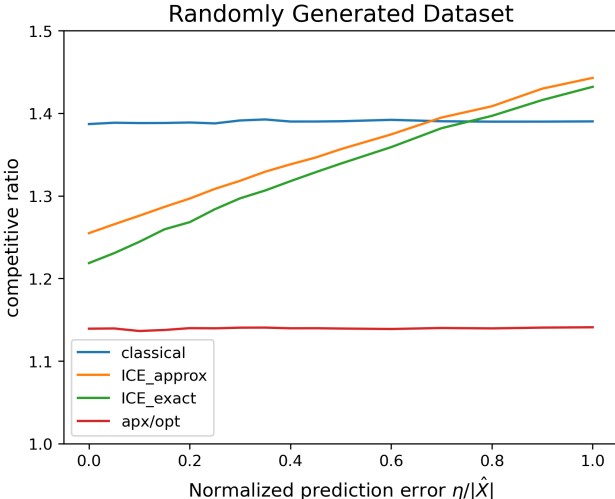

*Figure 1.* Competitive ratio for online set cover for varying prediction error $\eta$.

**Dataset** We use two datasets. The first one is the public dataset from the ongoing PACE Challenge, from the exact track on the Hitting Set Problem, (2) (this is equivalent to the Set Cover problem). On average, these instances have 2200 elements, and 1800 sets. The second dataset is randomly generated. Each instance consists of 1000 elements and 100 sets, each set contains 50 elements sampled uniformly from the groundset without replacement. Both datasets consist of 100 instances.

**Set-up and predictions** For every instance, we randomly fix $50\%$ of its elements to be included in the prediction $\hat{X}$. We obtain the real instances $X$ from $\hat{X}$ by replacing an $\alpha$-fraction of the predicted elements by unpredicted elements, keeping the total number of arriving elements fixed. This results in the normalized prediction error $\eta/|\hat{X}| = 2 \cdot \alpha$. For each obtained instance $X$, we report the multiplicative gap between the solution of the greedy algorithm and that of an IP-solver (apx/opt).

**Evaluation** We use the standard $O(\log k \log |\mathcal{S}|)$-competitive algorithm from (4) to instantiate ICE (Algorithm 2), and also use it as our baseline (classical). We compare the baseline to the (averaged) performance of our approach: we either use an exact decomposition for $\hat{X}$ (ICE_exact), corresponding to Theorem 1.1, or an approximate decomposition (ICE_approx), corresponding to Theorem 1.2. Whenever the baseline algorithm is about to make an arbitrary choice among several tied sets, we use the decomposition to break the tie. This does not affect our theoretical guarantees in any way. We achieve substantial improvements over the baseline, even for large prediction errors. The effects between using an exact or an approximate decomposition are negligible for the PACE dataset and quite small for the randomly generated instances, see Figure 1. Detailed statistics can be found in the full version and supplementary material.

## Impact Statement

This paper presents work whose goal is to advance the field of Machine Learning and Algorithm Design. There are many potential societal consequences of our work, none which we feel must be specifically highlighted here.

## Acknowledgements

A. Jabal Ameli was supported by the project COALESCE (ERC grant no. 853234).

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

# Appendix: Full Version

## 1. Introduction

In recent years, there has been significant work to incorporate *ML-advice* in the design of *online algorithms*. The basic premise is to provide additional information to an online algorithm to improve its guarantees. Ideally, whenever the prediction is correct, we expect to break through worst-case lower bounds, and the guarantees should degrade smoothly with the prediction error. This paradigm has been applied (quite successfully) to a wide array of online problems, ranging from caching (34; 6), scheduling (32; 33), graph algorithms (44; 9) and matching/secretary problems (19), to name a few. The rapid evolving of this research direction, often referred to as *learning-augmented algorithms*, is evidenced by the collection of papers in the dedicated website (1). Given the attention that the field has gained in the past years, designing general approaches for learning-augmented algorithms is of crucial interest for the community. In this paper, we contribute significantly to this goal by providing a new framework which applies to a general class of online *covering* problems.

Formally, a covering problem is given by some ground set $X'$, as well as a family of *covers* $\mathcal{S}$. Each $S \in \mathcal{S}$ indicates whether an element $e \in X'$ is covered by cover $S$ (hence, one can often identify a cover with a subset of $X'$). There is a cost function $\text{cost} : 2^{\mathcal{S}} \to \mathbb{R}_{\geq 0}$ associated to the covers, and the goal is to find the *cheapest* sub-family $\mathcal{S}' \subseteq \mathcal{S}$ of covers to cover all ground set $X'$. We make the following natural assumptions (that justify the term *covering*): if $\mathcal{S}^1 \subseteq \mathcal{S}$ covers $X^1 \subseteq X'$ and $\mathcal{S}^2 \subseteq \mathcal{S}$ covers $X^2 \subseteq X'$, then (i) $\mathcal{S}^1 \cup S^2$ covers $X^1 \cup X^2$, and (ii) $\text{cost}(\mathcal{S}' \cup \mathcal{S}'') \leq \text{cost}(\mathcal{S}') + \text{cost}(S'')$.

In the *online* setting considered in this paper, only a (unknown) subset of $X \subseteq X'$ requests needs to be actually covered. The elements of $X$ to be covered are revealed one-by-one, and we have to maintain a valid covering for the arrived elements in a monotone fashion (i.e. the decisions which sets to include are irrevocable). The goal is to minimize the *competitive ratio*. An online algorithm ALG for a *minimization* problem is $\rho$-competitive, if for any input $I$, the cost of the solution output by the online algorithm (denoted as $\text{cost}(\text{ALG}(I))$) is at most $\rho$ times the best solution for $I$ (denoted as $\text{cost}(\text{OPT}(I))$).

Many classical (online) optimization problems can be cast in the above form. Of course, the most natural one is the standard Set Cover problem. However, as we will discuss later, other examples include Facility Location problems, Steiner problems and Path Augmentation problems, among others. Most notably, these problems can be explicitly cast as instances of covering problems, though often with an exponential (in $|X|$) number of sets. Despite this, our method

is powerful enough to remain valid. The main condition we need is simply the covering property defined above: if a partial solution is feasible for the currently revealed elements (that corresponds, e.g., to a subset of demands or connectivity requests when considering facility location or graph problems), it remains feasible after adding new sets (e.g., facilities or edges) to accommodate newly arrived elements.

Our framework applies to the following natural *learning-augmented* setting for online covering problems. We assume that we receive upfront a prediction $\hat{X}$ on the elements that will eventually arrive. E.g. for the facility location problem, this prediction corresponds to arriving customers, or, for Steiner problems, to vertices that will eventually need to be connected to some root. This prediction is not necessarily completely accurate, and we define the *prediction error* $\eta$ as the symmetric difference between $\hat{X}$ and the set of actually arriving requests $X$, capped at $|X|$. Formally, $\eta := \min(|X|, |X \triangle \hat{X}|)$, where $X \triangle \hat{X} := (X \setminus \hat{X}) \cup (\hat{X} \setminus X)$. This is sometimes referred to as the $\ell_1$ error. We stress that our prediction error $\eta$ depends only on the *difference* between the set of actually arriving elements and the set of predicted elements. The actual ordering (in the arrival) is *not* important. In the following, we let $k := |X|$. Our framework shows that any online covering algorithm can be turned into a learning-augmented algorithm: provided there is an algorithm with competitive ratio depending on $k$, the number of actually arriving elements (e.g. $O(\log k \log |\mathcal{S}|)$ for set cover, $O(\log k)$ for Steiner tree), we obtain a learning-augmented online algorithm with a dependency on $\eta$, the prediction error ($O(\log \eta \log |\mathcal{S}|)$, $O(\log \eta)$ resp.). We emphasize that our approach operates in a fully black-box manner and does not rely on any particular feature of the online covering algorithm.

**Theorem 1.1** (Existence). *Let* ALG *be an algorithm for an online covering problem with a competitive ratio $\rho(k, \cdot)$, where $k$ is the number of arriving requests. Then, given a prediction $\hat{X}$, there exists an online algorithm with a competitive ratio of $O(\rho(\eta, \cdot))$.*

Our result demonstrates that for any online covering problem, it is always possible to incorporate predictions and obtain a refined competitive ratio that depends on the prediction error $\eta$. Furthermore, the resulting learning-augmented online algorithm is both *smooth* and *robust*: the competitive ratio $\rho(\eta, \cdot)$ improves over the baseline online algorithm whenever the prediction is (close to) perfect and smoothly degrades with the prediction error, but is never worse, even when the predictions are completely off (recall that $\eta \leq k$, by definition).

Our framework is based on a charging scheme that relies on a subtle *decomposition* of the solution space. Given some prediction $\hat{X}$, we compute *offline* partial solutions

$\mathcal{S}_1, \mathcal{S}_2, \ldots \subseteq \mathcal{S}$, such that their union is feasible for $\hat{X}$. In the online phase, we use a $\rho(k, \cdot)$-competitive algorithm to satisfy the requests in a black-box fashion. We only keep track of its expenses: whenever its cost exceeds that of some partial solution $\mathcal{S}_i$, we buy $\mathcal{S}_i$ (in addition to whatever the online algorithm buys). Interestingly, this simple framework turns out to be powerful enough to guarantee a competitive ratio that depends on the prediction error, rather than the actual request sequence.

We now turn to discussing the *efficiency*. In fact, whenever the decomposition mentioned above can be computed efficiently, then the overall online algorithm turns into a polynomial time algorithm. However, while such a decomposition always exists (yielding the existential result), it is not always possible to compute it in polynomial time. This is not surprising, as there are covering problems for which an $O(\rho(\eta, \cdot))$-competitive algorithm *cannot* be achieved in polynomial-time, assuming $\mathsf{P} \neq \mathsf{NP}$. An example of this is the classical Set Cover problem (see Theorem 3.3). Nevertheless, we are still able to provide a *relaxed* notion of decomposition, that we call an $(\alpha, \gamma)$-*decomposition* (details in Section 2). This relaxed notion allows us to smoothly 'compromise' between the competitive ratio and the efficiency of the overall framework. The resulting guarantees are resumed in the following theorem.

**Theorem 1.2** (Efficiency). *Let* ALG *be an algorithm for an online covering problem with a competitive ratio $\rho(k, \cdot)$, where $k$ is the number of arriving requests. Assume that an $(\alpha, \gamma)$-decomposition for the* offline *version of the covering problem can be computed in polynomial-time. Then, given a prediction $\hat{X}$, there exists a polynomial-time online algorithm with a competitive ratio of*

$$O(\alpha) \cdot \rho(\eta, \cdot) + O(\alpha \cdot \log^{-1}(\tfrac{\gamma}{\gamma-1}) \cdot \log k).$$

*In particular, if $\gamma = 1$, the competitive ratio becomes $O(\alpha \cdot \rho(\eta, \cdot))$.*

Note that a $(1, 1)$-decomposition always exists and yields Theorem 1.1. This *additive* term bypasses the aforementioned impossibility to turn any $\rho(k, \cdot)$-competitive algorithm into a learning-augmented, $O(\rho(\eta, \cdot))$-competitive algorithm in polynomial time.

We demonstrate the versatility of our approach, by applying it to the vast class of online covering problems. In particular, our framework yields the *first* learning-augmented algorithms when considering prediction on requests, for classical problems like set cover, non metric facility location, path augmentation, and hard generalizations of Steiner problems.

We conclude this introduction by giving an account of example applications (Section 1.1), and a comparison to related work (Section 1.2). Our framework is formally presented in Section 2, and the details for our example applications are discussed in Section 3. In Section 4 we present the experimental evaluation of our approach for the online set cover problem.

### 1.1. Applications

**Set Cover** An instance $(X', \mathcal{S}, w)$ of online set cover is given by a ground set $X'$, an ensemble of sets $\mathcal{S} \subseteq 2^{X'}$, as well as a cost function $w : \mathcal{S} \to \mathbb{R}_{\geq 0}$. A subset of the elements $e \in X \subseteq X'$ arrive one-by-one. The goal is to maintain a subcollection $\mathcal{S}' \subseteq S$ covering all $X$, while minimizing $\sum_{S \in \mathcal{S}'} w(S)$. Instances of Set Cover naturally translate into instances $(X', \mathcal{S}, \mathsf{cost})$ of our covering problem by letting $\mathsf{cost}(\mathcal{S}') = \sum_{S \in \mathcal{S}'} w(S)$ for all $\mathcal{S}' \subseteq \mathcal{S}$. Set Cover admits a *randomized* online algorithm with competitive ratio $O(\log k \log |\mathcal{S}|)$ (4). Our framework yields the first learning-augmented algorithm for predictions on request, with a $O(\log \eta \log |\mathcal{S}|)$-competitive ratio, plus an additive $O(\log k) \cdot \mathsf{OPT}$, if it is required to run in polynomial time.

**Facility Location** An instance of (non-metric) Facility Location (NMFL) is given by a client-facility graph $(C', F)$, opening cost $w : F \to \mathbb{R}_{\geq 0}$, assignment cost $\mathsf{dist} : C' \times F \to \mathbb{R}_{\geq 0} \cup \{+\infty\})$. Clients $c \in C \subseteq C'$ are released one-by-one. The goal is to maintain a set $F'$ of open facilities and an assignment $x : C \times F' \to \{0, 1\}$ of arrived clients to open facilities, minimizing $\sum_{f \in F'} w(f) + \sum_{c \in C, f \in F'} \mathsf{dist}(c, f) x(c, f)$. Instances of Facility Location translate into instances $(X', \mathcal{S}, \mathsf{cost})$ of our covering problem by letting $X' = C'$, and by defining a set $S \in \mathcal{S}$ for every possible subset $F' \subseteq F$ and every possible assignment $\tilde{x}$ of a subset $\tilde{C}$ of clients to $F'$. The $\mathsf{cost}(S)$ of $S$ would be $\sum_{f \in F'} w(f) + \sum_{c \in \tilde{C}, f \in F'} \mathsf{dist}(c, f) \tilde{x}(c, f)$, and for any $\mathcal{S}' \subseteq \mathcal{S}$, $\mathsf{cost}(\mathcal{S}') = \sum_{S \in \mathcal{S}'} \mathsf{cost}(S)$. For this problem, there exists a *randomized* online algorithm with competitive ratio $O(\log k \log |F|)$ (4). Our framework yields a $O(\log \eta \log |F|)$- competitive algorithm, plus an additive $O(\log k) \cdot \mathsf{OPT}$, if it is required to run in polynomial time. To the best of our knowledge, our algorithm establishes the first learning-augmented algorithm under prediction on request for Facility Location. Previously, only the *metric* variant (that is, when the assignment function (dist) is a metric function) was studied (9). For this setting, the best competitive ratio is $O(\frac{\log k}{\log \log k})$, (35; 20). Our framework yields a polynomial-time, $O(\frac{\log \eta}{\log \log \eta})$-competitive learning-augmented algorithm. We elaborate in Section 1.2 how this compares with the result of (9), as their result holds with respect to a slightly different error measure, and most importantly, cannot be applied in a complete black-box fashion.

**Parking Permit and Weighted Path Augmentation** An instance of weighted path augmentation (WPAP) is given by $(E, \mathcal{L}, \mathsf{cost} : \mathcal{L} \to \mathbb{R}_{\geq 0})$. $E$ is a set of elements, that

correspond to edges of a path $P$, indexed from 1 to $n$, and $\mathcal{L}$ is a set of *links*, weighted according to a cost function. Each link covers a consecutive interval of elements (subpath of $P$). In the online phase, elements arrive one-by-one, and we need to immediately and irrevocably select a link covering it. This can be thought as a connectivity augmentation problem where edges of a path can fail online, and we need to restore connectivity. Instances of this problem map into instances $(X', \mathcal{S}, \mathsf{cost})$ of our covering problem by letting $X' = E, \mathcal{S} = \mathcal{L}$. The Parking Permit problem is a special case of WPAP. For any element (edge of the path), we can select $K$ different permits (links) of fixed durations (i.e. length). For both problems, it is possible to devise a deterministic algorithm that is $O(\log k)$-competitive, as well as a randomized online algorithm that is $O(\log \log k)$-competitive against oblivious adversaries. Our results yield a deterministic, $O(\log \eta)$-competitive online algorithm and a randomized, $O(\log \log \eta)$-competitive online algorithm. Again, these are the first learning-augmented algorithms for connectivity augmentation problems. We provide the details in Section 3.2.

**Steiner Tree and variants** In Steiner tree (resp. forest) problems, given a metric graph $G$, terminals $\{t_1, t_2, \ldots, t_k\}$ show up one-by-one (resp. in pairs), and one needs to maintain a connected subgraph connecting all arrived terminals (terminal pairs). Instances of this problem map into instances $(X', \mathcal{S}, \mathsf{cost})$ of our covering problem by letting $X' = V$, $\mathcal{S}$ be the set of all possible $t_i - t_j$ paths in the graph, and by defining $\mathsf{cost}(\mathcal{S}')$ to be the sum of edge (or nodes) weights in the union of the corresponding paths. In the edge-weighted setting, there are $O(\log \eta)$-competitive algorithms for these two problems (44; 9), which we can also obtain with our framework. However, for several important variations such as group Steiner tree / forest, as well as the node-weighted and directed setting, no learning-augmented algorithms were known prior to our work. For these problems, the competitive ratio is of order $O(\log^{O(1)} k \log^{O(1)} |V|)$, (37; 16). Our framework directly improves this to $O(\log^{O(1)} \eta \log^{O(1)} |V|)$ in the learning-augmented setting. We provide the details in Section 3.3.

### 1.2. Comparison to related work and techniques

The study of general schemes in the context of learning-augmented online algorithms for set cover (and other covering problems) was initiated by (11). They devise a primal-dual approach for covering problems that incorporates predictions. Subsequent works on the problem then consider further variations, see for instance (5; 24; 10). However, a common feature of these works, is that the prediction is on the sets, i.e. the online algorithms receives hints on what an optimal solution (conditioned on the set of actually arriving elements) looks like. As such, these results are incomparable to ours.

Despite this, we would like to mention that the results in (11; 23; 10) can still be *combined* with our framework. Informally, given a prediction on sets in an optimal solution, they provide an algorithm that maintains a $f(\mathsf{Qual}_{\mathsf{pred}})$-competitive *fractional* solution. $\mathsf{Qual}_{\mathsf{pred}}$ is the *quality* of the prediction (with respect to some error metric), and $f(\mathsf{Qual}_{\mathsf{pred}}) \leq O(\log |\mathcal{S}|)$. To obtain an integral solution, they round it in an online fashion. This incurs a multiplicative loss of $O(\log k)$. Hence, combined with our framework, this yields a polynomial-time $O(\log \eta \cdot f(\mathsf{Qual}_{\mathsf{pred}}) + \log k)$-competitive algorithm for online set cover that incorporates predictions on arriving requests, as well as sets of an optimal solution.

A general framework that is close to ours is the one by (9). Here they study an interesting generalization of the error we consider, specialized to problems on metric graphs. Specifically, they consider the *metric error with outliers* given by $(\triangle, D)$: for some $\hat{T} \subseteq \hat{X}$, $T \subseteq X$ with $|T| = |\hat{T}|$, and a min-cost matching between $T$ and $\hat{T}$ of cost $D$, and $|\hat{X} \setminus \hat{T} \cup X \setminus T| = \triangle$. An important feature of their work, is that their guarantees hold with respect to the best choice of $\triangle$ and $D$. In particular, setting $\hat{T} = T = \hat{X} \cap X$ (consequently $D = 0$), their error measure is always smaller than $\eta$ for metric graph problems. On a high-level, the overall approach is similar as they also compute partial solutions to the previously arrived elements, and pay towards predicted requests. This is decided based on an instance of a *prize-collecting* variant of the problem. However, to obtain their guarantees, they require the underlying online algorithm to be *subset-competitive*, a non-trivial property (for instance for some online algorithms for the metric facility location problem). In contrast, we rely on a *more involved* decomposition, but our results *do not* depend on specifics of the underlying online algorithm and can be applied in a completely modular and black-box way. As a direct consequence, we can also apply our results to any standard algorithm for these problems. In particular, by leveraging the *best* known online algorithm for metric facility location with competitive ratio $O(\frac{\log k}{\log \log k})$, we obtain a bound of $O(\frac{\log \eta}{\log \log \eta})$. In comparison, they give a $O(\log \triangle)$-competitive ratio (plus an additive term of $D$), as they rely on a $O(\log k)$-competitive algorithm to ensure subset-competitiveness. As a side note, we remark that there has been significant work on learning-augmented online facility location, see (5; 3; 29; 10; 21). However, all these works assume that predictions are hints on the location of optimal facilities. As such, these results are incomparable to ours. The framework in (9) also applies to Steiner tree and Steiner forest, yielding a $O(\log \triangle)$-(plus additive $D$) competitive ratio (a bound of $O(\log \eta)$ for Steiner tree was previously given in (44)). Our framework recovers this latter guarantee of $O(\log \eta)$, but is also applicable for the generalisations in Section 1.1.

For the parking permit and its generalization, the weighted

path augmentation problem, there were no known learning-augmented algorithms. The parking permit was introduced in (36), where a deterministic $O(\log n)$-, and a randomized, $O(\log \log n)$-competitive algorithm was given. However, it is straightforward to adapt this algorithm to yield a $O(\log k)$-(resp. $O(\log \log k)$-) competitive algorithm for parking permit. A generalisation of WPAP was first studied by (38), and they provide a deterministic, $O(\log n)$ competitive algorithm for the problem, as well as a fractional, $O(\log \log n)$-competitive algorithm for WPAP. We adapt these ideas to yield a deterministic, $O(\log k)$-competitive algorithm for WPAP, as well as a randomized, $O(\log \log k)$-competitive algorithm, for which our framework can be applied.

We conclude this subsection with a final remark. The error $\eta$ we consider is essentially the $\ell_1$ error, the term $\min\{|X|, \cdot\}$ simply follows from robustification. As such, our error measure is very simple and efficient to learn. We briefly sketch this. Consider the setting where we are given a distribution over instances of some covering problem with requests in $X'$. The goal is to find a "best-possible" predictor of the form $\hat{x} \in \{0, 1\}^{|X'|}$, which minimizes the expected $\ell_1$-distance (e.g. $\eta$) to a request sequence $x \in \{0, 1\}^{|X'|}$ sampled from the distribution. Note that by rounding, we can indeed assume $\hat{x}$ to be integral. Since the $\ell_1$-norm is just a sum over all components, it suffices to minimize the expected Hamming distance for each component separately. For any such component, taking the majority over $O(\log |X'|)$ i.i.d. samples, ensures that with probability at least $1 - 1/\mathsf{poly}(|X'|)$, the resulting $\{0, 1\}$-majority vote is within $0.49$ of the true predictor $\hat{x}$. Taking the union bound over all $|X'|$ ensures correctness for all components (with high probability). For a textbook argument, see e.g. Chapter 4 of (41). Finally, we note that the error measure $\eta$ also naturally handles fractional predictions, where each request is associated with some probability or confidence parameter. Again, this follows from the fact that we can always round such a prediction (round up each component if it is larger than $1/2$, round down otherwise) while preserving the (expected) error.

## 2. Framework for predictions on requests

We now describe ICE (**I**teratively **C**harge **E**xpenses), a deterministic online procedure to incorporate predictions on requests. This procedure operates in two stages. In the first, offline phase, we receive the prediction $\hat{X}$ on the requests. We compute a decomposition for $\hat{X}$, i.e.

$$\hat{X}_1 \sqcup \hat{X}_2 \sqcup \ldots \sqcup \hat{X}_n = \hat{X},$$

and corresponding feasible solutions $\mathcal{S}_1, \mathcal{S}_2, \ldots, \mathcal{S}_n$, of respective costs $c(\mathcal{S}_i)$. In the online phase, we keep track of the cost by ALG incurred on elements in $X \cap \hat{X}$. Whenever this exceeds the cost of $\mathcal{S}_i$, we buy $\mathcal{S}_i$. We resume this in Algorithm 2.

---

**Algorithm 2** ICE - Iteratively Charge Expenses
1: **Input:** Online algorithm ALG with competitive ratio $\rho(k, \cdot)$ and predicted requests $\hat{X}$.
2: **Offline phase:** Compute $(X_i, \mathcal{S}_i)$s as detailed in Section 2.1.
3: Initialize two instances of ALG, $\mathsf{ALG}_+$ and $\mathsf{ALG}_-$.
4: Set excess $\leftarrow 0$, layer $\leftarrow 1$.
5: **when** request $x \in X$ arrives
6:   **if** $x \notin \hat{X}$ **then**
7:     Pass $x$ to $\mathsf{ALG}_+$.
8:   **else**
9:     Pass $x$ to $\mathsf{ALG}_-$. Increase excess by the cost of the partial solution $\mathsf{ALG}_-$ selects for $x$.
10:     **while** excess $\geq c(\mathcal{S}_{\mathsf{layer}})$ **do**
11:       Buy $S_{\mathsf{layer}}$, layer $\leftarrow$ layer $+ 1$, excess $\leftarrow$ excess $- c(\mathcal{S}_{\mathsf{layer}})$ and reinitialize $\mathsf{ALG}_-$.
12:     **end while**
13:   **end if**
14: **end when**

---

The running time of our procedure is at most that of the online algorithm that it uses in a black-box manner, as well as the time to compute the decomposition in Line 2. We formalize this in the following subsection.

### 2.1. Offline phase of the General Framework

We first illustrate our approach. To this end, assume that we know the exact value of $\eta$ and that $X \subseteq \hat{X}$.[2] Our framework can be simplified to the following two steps: in the preliminary (offline) stage, we compute $\tilde{S}$, a minimum cost collection of sets, that covers $|X| - \eta$ elements of $X$. In the online part of the algorithm, we use the $\rho(k, \cdot)$-competitive algorithm in a black-box fashion. Once the cost of selected sets exceeds $\tilde{S}$, we buy all sets belonging to $\mathcal{S}$. To see why this results in an $O(\rho(\eta, \cdot))$-competitive algorithm, observe that the cardinality of the elements in $X$ that are not covered by a set in $\tilde{S}$, is at most $\eta$. Hence on these $\eta$ elements, the online algorithm is $O(\rho(\eta, \cdot))$-competitive. On the other hand, on the elements covered by $\tilde{S}$, the online algorithm can never spend more than OPT: this is since the cost of $\tilde{S}$ is a lower bound on the optimum, and we buy all sets belonging to $\tilde{S}$ once the online algorithm spends more than the cost of $\tilde{S}$.

Unfortunately, it seems there is no direct way to obtain an estimate on $\eta$, standard techniques like guess-and-double do not work. Instead, we take a different route. We construct sets $\mathcal{S}_i$, whose union progressively gets 'closer' to the (unknown) set $\tilde{S}$. Indeed, it is not necessary to exactly find $\tilde{S}$, but is sufficient to approximate it in terms of cardinality

---

[2]There are at most $\eta$ elements in $X \setminus \hat{X}$, so we can use a separate online algorithm for these elements that is $\rho(\eta, \cdot)$-competitive.

and total cost. To handle both, we impose some geometric growth condition on the sets $\mathcal{S}_i$, see properties (A) and (B) below. Each $\mathcal{S}_i$, analogously to $\tilde{\mathcal{S}}$, is the cheapest collection of sets covering some prescribed fraction of (remaining) elements. Computing these sets is typically an NP-hard problem, which requires approximation to remain tractable. To this end, we introduce the notion of $(\alpha, \gamma)$-decomposability, see Definition 2.1, which we use in Properties (C) and (D) of the decomposition to control the total cost of these sets $\mathcal{S}_i$, and bound the resulting competitive ratio.

**Definition 2.1** (Efficient $(\alpha, \gamma)$-decomposable)**.** An offline covering problem with request set $X$ is said to be $(\alpha, \gamma)$-decomposable if for all $i \in \{1, 2, \ldots, |X|\}$, we can compute a set $B \subseteq X$ and a collection $\mathcal{S}_B$ covering $B$, s.t.

- The cost of $\mathcal{S}_B$ is at most $\alpha$ times the minimum cost of covering $i$ requests in $X$, i.e.

$$c(\mathcal{S}_B) \leq \alpha \cdot \min_{A \subseteq X, |A|=i} \{c(\mathcal{S}_A) \mid \mathcal{S}_A \text{ covers } A\}, \text{ and,}$$

- $|B| \geq i/\gamma$.

Whenever $B$ and $\mathcal{S}_B$ can be computed in polynomial time, we say the problem is *efficiently* $(\alpha, \gamma)$-decomposable.

Note that any problem is $(1, 1)$-decomposable, e.g. through integer programming or enumeration techniques. However, this may take super-polynomial time, as is the case, e.g., for set cover.

Next, we use this property of $(\alpha, \gamma)$-decomposability, to obtain an $(\alpha, \gamma)$-*decomposition*. For convenience, we set $g_{\alpha,\gamma}(t) := \alpha$ if $\gamma = 1$ and $g_{\alpha,\gamma}(t) := \alpha \cdot (1 + \log^{-1}(\frac{\gamma}{\gamma-1}) \cdot \log t)$ if $\gamma > 1$.

We split $\hat{X}$ through an iterative process: starting with $R_0 := \hat{X}$, we select a subset $X_1 \subseteq R_0$, update $R_1 \leftarrow R_0 \setminus X_1$, and recurse with $R_1$. Then for every $i$, we denote by $\mathcal{S}_i$ the ($\alpha$-approximate) solution computed to cover the elements in $X_i$, as in Definition 2.1. An $(\alpha, \gamma)$-decomposition is then a pair $(X_i, \mathcal{S}_i)$, where $\hat{X} = X_1 \cup X_2 \cup \cdots \cup X_r$, and the following properties hold for each $i$:

(A) $|X_i| \geq \frac{|\hat{X}| - (|X_1| + \ldots + |X_{i-1}|)}{2}$.

(B) If $c(\mathcal{S}_i) < 2c(\mathcal{S}_{i-1})$, then $8c(\mathcal{S}_{i-1}) < c(\mathcal{S}_{i+1})$.

(C) If $c(\mathcal{S}_i) > 10c(\mathcal{S}_{i-1})$, then $c(\mathcal{S}_i) \leq g_{\alpha,\gamma}(|\hat{X}|) \cdot C_1$, where $C_1$ is the minimum cost solution that covers $\lceil \frac{|\hat{X}| - (|X_1| + \ldots + |X_{i-1}|)}{2} \rceil$ elements from $R_{i-1}$.

(D) $c(\mathcal{S}_i) \leq g_{\alpha,\gamma}(|\hat{X}|) \cdot C_2$, where $C_2$ is defined as follows: for $i \geq 2$, $C_2$ is the minimum cost solution that covers $|X_i|$ elements from $R_{i-1}$; for $i = 1$, $C_2$ is the minimum cost solution that covers $\lceil |\hat{X}|/2 \rceil$ elements from $R_0$.

**Theorem 2.2.** *There is a polynomial-time construction guaranteeing properties (A), (B), (C), and (D), provided that the problem is efficiently $(\alpha, \gamma)$-decomposable.*

To prove it we first need the following useful lemma.

**Lemma 2.3.** *If the problem is efficiently $(\alpha, \gamma)$-decomposable, a $g_{\alpha,\gamma}(|\hat{X}|)$-approximation for computing a minimum cost solution to cover at least $i$ elements from $\hat{X}$ can be computed in polynomial time.*

*Proof.* We iteratively apply the following process. Start with $h = i$, $F := \hat{X}$ and $S := \emptyset$. At each step compute a solution $S'$ in polynomial time, that covers at least $\frac{h}{\gamma}$ elements whose cost is at most $\alpha$ times the cost of the minimum cost solution that covers a subset of size $h$ from $F$. $S'$ covers a subset $F'$ of $F$, with $|F'| \geq h/\gamma$. We update $S \leftarrow S \cup S'$, $F \leftarrow F \setminus F'$, and $h \leftarrow h - |F'|$. We stop as soon as $h \leq 0$.

When the process terminates $S$ contains a solution that covers at least $i$ elements. The number of steps required is one if $\gamma = 1$ and otherwise is upperbounded by $\log_{1-1/\gamma} i + 1$ as we cover at least $h/\gamma$ many elements.

Also the solution $S'$ computed at each stage is an $\alpha$-approximation for minimum cost solution to cover at least $i$ elements from $\hat{X}$. Thus, if $\gamma = 1$, this is an $\alpha$-approximation and otherwise it is an $(\alpha \cdot (1 + \log^{-1}(\frac{\gamma}{\gamma-1}) \cdot \log i))$-approximation. The claim follows as $i \leq |\hat{X}|$. $\square$

We are now going to detail the construction. $\mathcal{S}_1$ is a solution that covers a subset $X_1 \subseteq R_0$ such that $|X_1| \geq \lceil |R_0|/2 \rceil = \lceil |\hat{X}/2| \rceil$ many elements of $R_0 = \hat{X}$, such that its cost is at most $g_{\alpha,\gamma}(\hat{X})$ times the cost of the optimal set that covers these many elements from $R_0$. Observe that $X_1$ and $\mathcal{S}_1$ can be simply computed by Lemma 2.3 and hence it satisfies Properties (A) and (D) for $i = 1$.

Now for every $i \geq 1$, we first compute $R_i$, by simply setting $R_i \leftarrow R_{i-1} \setminus X_i$. Now we show how we compute $\mathcal{S}_{i+1}$ and $X_{i+1}$.

First, for every $j \geq \lceil \frac{|R_i|}{2} \rceil$ we find a subset $X_{i+1,j} \subseteq R_i$ of elements, together with a feasible solution $\mathcal{S}_{i+1,j}$ to cover it, such that:

i) $|X_{i+1,j}| \geq j$,

ii) $\mathcal{S}_{i+1,j}$ is an $g_{\alpha,\gamma}(|\hat{X}|)$-approximation for the minimum cost to cover among all subsets of $R_i$ of cardinality $j$,

iii) $c(\mathcal{S}_{i+1,j}) \leq c(\mathcal{S}_{i+1,j+1})$,

iv) If $\mathcal{S}_{i+1,j}$ covers more than $j$ elements of $R_i$ then $c(\mathcal{S}_{i+1,j+1}) = c(S_{i+1,j})$,

v) and for every $e \in R_i \setminus X_{i+1,j}$, it holds that $c_e + c(\mathcal{S}_{i+1,j}) \geq c(\mathcal{S}_{i+1,j+1})$, where $c_e$ is the cost of the cheapest set that covers $e$.

For computing $X_{i+1,j}$s and $\mathcal{S}_{i+1,j}$s that satisfy i) to v) we first start with computing $X_{i+1,j}$s and $\mathcal{S}_{i+1,j}$s that satisfy i) and ii), using Lemma 2.3 by replacing $\hat{X}$ by $R_i$ and replacing $i$ by $j$ in this lemma. Now, if for any $j$, $c(\mathcal{S}_{i+1,j}) > c(\mathcal{S}_{i+1,j+1})$, we update $X_{i+1,j} \leftarrow X_{i+1,j+1}$ and $\mathcal{S}_{i+1,j} \leftarrow \mathcal{S}_{i+1,j+1}$. The updated $X_{i+1,j}$s and $\mathcal{S}_{i+1,j}$s satisfy i), ii), and iii).

Now assume that iv) or v) does not hold for some $j$; Let $j^*$ be the smallest $j$ that has this property. If iv) does not hold for $j^*$, then we simply update $\mathcal{S}_{i+1,j^*+1} \leftarrow \mathcal{S}_{i+1,j^*}$ and we update $X_{i+1,j^*+1}$ to the set of all elements in $R_i$ that are covered by $\mathcal{S}_{i+1,j^*}$. Otherwise, if v) does not hold for $j^*$, then we take an element $e \in R_i \setminus X_{i+1,j^*}$, where $c_e$ is minimum and $c_e + c(\mathcal{S}_{i+1,j}) < c(\mathcal{S}_{i+1,j^*+1})$. We update $X_{i+1,j^*+1} \leftarrow X_{i+1,j^*} \cup (S_e \cap R_i)$ and $\mathcal{S}_{i+1,j^*+1} \leftarrow \mathcal{S}_{i+1,j^*} \cup \{S_e\}$, where $S_e$ is the cheapest set that covers $e$.

Note that after these modifications i), ii), and iii) are still satisfied and iv) and v) are still satisfied for any $j \leq j^*$. Also properties iv and v) are now ensured for $j^*$ as well. Hence by repeating this process we can ensure i) to v) for every $j$. Now using $X_{i+1,j}$s and $\mathcal{S}_{i+1,j}$s, we use the following rules to construct $X_{i+1}$ and $\mathcal{S}_{i+1}$:

**Case 1**: If $c(\mathcal{S}_{i+1,\lceil|R_i|/2\rceil}) \geq 2c(\mathcal{S}_i)$, we set $X_{i+1} \leftarrow X_{i+1,\lceil|R_i|/2\rceil}$ and $\mathcal{S}_{i+1} \leftarrow \mathcal{S}_{i+1,\lceil|R_i|/2\rceil}$.

**Case 2**: If not, let $\ell$ be the largest number such that $c(\mathcal{S}_{i+1,\ell}) \leq 10c(\mathcal{S}_i)$ (Clearly $\ell \geq \lceil\frac{|R_i|}{2}\rceil$). In this case we set $\mathcal{S}_{i+1} \leftarrow \mathcal{S}_{i+1,\ell}$ and $X_{i+1} \leftarrow X_{i+1,\ell}$.

Now we have all the ingredients required to prove Theorem 2.2

*Proof of Theorem 2.2.* It suffices to show that the above construction yields properties (A), (B), (C), and (D). Properties (A) and (D) are clearly achieved by construction and by i) to v).

Property (C) is also achieved by construction, as if $10c(\mathcal{S}_i) < c(\mathcal{S}_{i+1})$ then in our construction we must be in **Case 1**. Now in **Case 1**, $\mathcal{S}_{i+1}$ is simply $\mathcal{S}_{i+1,\lceil|R_i|/2\rceil}$, which is a $g_{\alpha,\gamma}(\hat{X})$-approximation for a solution that covers $\lceil|R_i|/2\rceil$ many elements of $R_i$, and hence the claim.

Now we show that Property (B) holds; To prove this, note that in our construction if $c(\mathcal{S}_i) < 2c(\mathcal{S}_{i-1})$ then we should be in **Case 2** at stage $i$. However, in **Case 2**, by our choice of $\ell$, the cost to cover any additional element (i.e. any element in $R_i$) should be at least $8c(\mathcal{S}_{i-1})$; as otherwise, by v), $\ell$ is not the largest possible number as described in **Case 2**, a contradiction. Therefore, $c(\mathcal{S}_{i+1}) \geq 8c(\mathcal{S}_{i-1})$ □

## 2.2. General Framework Analysis

In this section, we analyze the competitive ratio of Algorithm 2. Assume that upon the termination of our algorithm (Algorithm 2), we are at layer $i+1$ and $\overline{\text{excess}}$ is the value of excess when the algorithm ends. That is, in the previous steps we were able to buy $\mathcal{S}_1 \cup \ldots \cup \mathcal{S}_i$, and $\text{layer} = i+1$ when the algorithm terminates, with $\overline{\text{excess}} < c(\mathcal{S}_{i+1})$. We also define $\Delta_- := |\hat{X} \setminus X|$.

The following two lemmata give a lower bound on the size of the optimal solution.

**Lemma 2.4.** *Let $m$ be a positive integer. If $\Delta_- \leq |R_m|$, then $\mathsf{OPT} \geq \max_{1 \leq j \leq m}\{c(\mathcal{S}_j)\}/g_{\alpha,\gamma}(|\hat{X}|)$.*

*Proof.* Let $j \in \{1, \ldots, m\}$. As $\Delta_- \leq |R_m|$, then $\Delta_- \leq |R_j|$. This means that at least $|X_j|$ many elements from $R_{j-1}$ appear in our instance (otherwise $\Delta_- > |R_j|$ as $|R_j| + |X_j| = |R_{j-1}|$). So, by Property (D), the cost to cover these elements is at least $c(\mathcal{S}_j)/g_{\alpha,\gamma}(|\hat{X}|)$. □

**Lemma 2.5.** *The following holds:*

- *For every $j \in \{1, \ldots, i\}$, $\mathsf{OPT} \in \Omega(c(\mathcal{S}_j)/\rho(\Delta_-,.))$ or $\mathsf{OPT} \in \Omega(c(\mathcal{S}_j)/g_{\alpha,\gamma}(|\hat{X}|))$, and*

- *$\mathsf{OPT} \in \Omega(\overline{\text{excess}}/\rho(\Delta_-,\cdot)$ or $\mathsf{OPT} \in \Omega(\overline{\text{excess}}/g_{\alpha,\gamma}(|\hat{X}|))$.*

*Proof.* For every $j \in \{1, \ldots, i+1\}$, let us denote by $a_j$ the minimum among the value of excess at the beginning of layer $j$ and $c(\mathcal{S}_j)$. Also for every $j \in \{1, \ldots, i\}$ we define $c(\text{ALG}_j) := c(\mathcal{S}_j) - a_j$. For $j = i+1$, we define $c(\text{ALG}_j) := \overline{\text{excess}} - a_j$. Consequently, we have $a_1 = 0$, and $0 \leq a_j \leq c(\mathcal{S}_j)$.

To better explain the above notation, consider the following scenario; Assume we are at $\text{layer} = j$, and then an element $x \in \hat{X}$ arrives. According to line 9 of Algorithm 2, ALG buys the partial solution $S$ for $x$. Now we update $\text{excess} \leftarrow \text{excess} + c(S)$. Now if $\text{excess} \geq c(\mathcal{S}_{\text{layer}})$, let $z$ be the largest number such that $\text{excess} \geq \sum_{i=\text{layer}}^{z} c(\mathcal{S}_i)$. In this case, we will have $a_{j+1} = c(\mathcal{S}_{j+1}), a_{j+2} = c(\mathcal{S}_{j+2}), \ldots, a_z = c(\mathcal{S}_z)$, and $c(\text{ALG}_{j+1}) = c(\text{ALG}_{j+2}) = \cdots = c(\text{ALG}_z) = 0$; Also we will have $a_{z+1} = \sum_{i=\text{layer}}^{z} \text{excess} - c(\mathcal{S}_i)$.

To prove the claim for any $j$, we will show that $\mathsf{OPT} \in \Omega(\min\{\frac{c(\text{ALG}_j)}{\rho(\Delta_-,\cdot)}, \frac{c(\text{ALG}_j)}{g_{\alpha,\gamma}(|\hat{X}|)}\} + a_j)$.

We first show that $\mathsf{OPT} \geq a_j$. As $a_1 = 0$, the claim is trivial for $j = 1$. If $j \geq 2$ the cost that the online algorithm spends at the stage which led to purchasing $\mathcal{S}_{j-1}$ is at least $a_j$, and therefore $a_j$ is a lower bound on $\mathsf{OPT}$; This is true as without loss of generality, ALG never pays more than $\mathsf{OPT}$ for any element.

To see this, note that the cheapest set covering any arriving element has cost at most OPT. We can always buy such a set, and pay the same amount towards buying the set proposed by the online algorithm. The overall cost is at most twice that of the online algorithm. Alternatively, for any online algorithm, a guess on (a 2-approximation of) OPT can be maintained online (again losing a factor 2 in the final guarantee). Hence, all sets of cost larger than this guess need not be considered.

To complete the argument, we will show $\mathsf{OPT} \in \Omega(\frac{c(\mathrm{ALG}_j)}{\max\{\rho(\Delta_-,\cdot),g_{\alpha,\gamma}(|\hat{X}|)\}})$. Note that the number of elements handled by our algorithm in layer $j$ is at most $|X_j|$, as otherwise by Property (D), the cost to cover only these elements is at least $\mathsf{OPT} \geq c(\mathrm{ALG}_j)/g_{\alpha,\gamma}(|\hat{X}|)$.

So by the assumption on the competitive ratio of ALG we have $\mathsf{OPT} \in \Omega(\frac{c(\mathrm{ALG}_j)}{\rho(|X_j|,\cdot)})$ (the online algorithm for elements that arrive at layer $j$, is an instance with at most $|X_j|$ elements and hence is $\rho(|X_j|,\cdot)$-competitive). This means that if $\Delta_- \geq |X_j|/2$ we are done as $\rho(\Delta_-,\cdot) \in \Omega(\rho(|X_j|,\cdot))$ (note that the competitive ratio can grow at most linearly). So we assume that $\Delta_- < |X_j|/2$. As $X_j \subseteq R_{j-1}$, then $\Delta_- \leq |R_{j-1}|$ and by Lemma 2.4 if $j > 1$, we have $\mathsf{OPT} \geq c(\mathcal{S}_{j-1})/g_{\alpha,\gamma}(|\hat{X}|)$.

We distinguish two cases; If $j > 1$ and $c(\mathcal{S}_j) \leq 10c(\mathcal{S}_{j-1})$, then $\mathsf{OPT} \geq c(\mathcal{S}_{j-1})/g_{\alpha,\gamma}(|\hat{X}|)$ already implies $\mathsf{OPT} \in \Omega(c(\mathrm{ALG}_j)/g_{\alpha,\gamma}(|\hat{X}|))$, and hence the claim. Now consider the case $j = 1$ and $c(\mathcal{S}_j) > 10c(\mathcal{S}_{j-1})$; As $\Delta_- < |X_j|/2$, we have $\Delta_- < |R_{j-1}|/2$. By Property (C) for $j > 1$ and Property (D) for $j = 1$, it holds that $\mathsf{OPT} \geq c(\mathcal{S}_j)/g_{\alpha,\gamma}(|\hat{X}|)$. $\square$

In the next claim, we give an upper-bound on the cost of ALG$_-$.

**Claim 2.1.** *The cost of* ALG$_-$ *can be upper bounded by* $O(c(\mathcal{S}_{i-1}) + c(\mathcal{S}_i)) +$ excess.

*Proof.* The cost of ALG$_-$ is $O(\sum_{j=1}^i c(\mathcal{S}_j)) +$ excess. By Property (B), for every $j$, $c(\mathcal{S}_j) + c(\mathcal{S}_{j+1}) \leq \frac{3}{4}(c(\mathcal{S}_{j+2}) + c(\mathcal{S}_{j+3}))$. So $\sum_{j=1}^{i-2} c(\mathcal{S}_j) \in O(c(\mathcal{S}_{i-1}) + c(\mathcal{S}_i))$ and hence the claim. $\square$

Before proving our main result of this section, namely Theorem 2.7, we introduce some useful notation. We define $k_- : |X \cap \hat{X}|$ and $k_+ := |X \setminus \hat{X}|$ as the number of arriving elements that are handled by ALG$_-$ and ALG$_+$, respectively ($k_- + k_+ = k$). We first analyze the competitive ratio of ALG$_-$.

**Lemma 2.6.** *Our algorithm for* $ALG_-$ *is* $O(\rho(\eta,\cdot) + g_{\alpha,\gamma}(k_-))$-*competitive.*

*Proof.* We first assume $k_- \geq |\frac{\hat{X}}{2}|$. From Lemma 2.5 and Claim 2.1, we observe that our algorithm is $O(\rho(\Delta_-,\cdot) + g_{\alpha,\gamma}(|\hat{X}|))$-competitive. As $k_- \geq \frac{|\hat{X}|}{2}$, and by definition of $\Delta_-$, $\eta \geq \min\{k_-,\Delta_-\}$, and therefore $\rho(\eta,\cdot) + g_{\alpha,\gamma}(|\hat{X}|) \in O(\rho(\eta,\cdot) + g_{\alpha,\gamma}(k_-))$.

Now assume $k_- < |\hat{X}|/2$. In this case $k_- < \eta$, and it suffices to show that our algorithm is $O(\rho(k_-,.))$-competitive. For this purpose, we observe that at each layer $j$ of ALG$_-$ our cost is at most $O(\rho(k_-,.))\mathsf{OPT}$, as at most $k_-$ elements arrive. By Claim 2.1 our algorithm is $O(\rho(k_-,.))$-competitive. $\square$

Now we have all the ingredients to analyze the competitive ratio of Algorithm 2.

**Theorem 2.7.** *Algorithm 2 is* $O(\rho(\eta,\cdot) + g_{\alpha,\gamma}(k))$-*competitive.*

*Proof.* ALG$_+$ is $O(\rho(k_+,\cdot))$-competitive as it is an instance of ALG with $k_+$ arriving elements. By Lemma 2.6, ALG$_-$ is $O(\rho(\eta,\cdot) + g_{\alpha,\gamma}(k_-))$-competitive. By definition of $k_+$, it holds that $k_+ \leq \eta$ and hence $\rho(k_+,\cdot) \in O(\rho(\eta,\cdot))$. Therefore, Algorithm 2 is $O(\rho(\min\{\eta,k\},\cdot) + g_{\alpha,\gamma}(k))$-competitive. $\square$

Theorems 1.1 and 1.2 follow from Theorems 2.7 and 2.2.

## 3. Applications

### 3.1. Set Cover and Non-Metric Facility Location

The main theorem of this section is the following.

**Theorem 3.1.** *There is a polynomial-time, $O(\log \eta \log |\mathcal{S}| + \log k)$- (resp. $O(\log \eta \log |F| + \log k)$-) competitive algorithm for online set cover (resp. non-metric facility location). In exponential time, the dependence on $\log k$ can be removed.*

By Theorem 2.2 combined with Theorem 1.2, it is sufficient to show that these problems are efficiently $(\alpha,\gamma)$-decomposable. This is shown in Lemma 3.2. For the exponential-time variant, we use Theorem 1.1 (observe once again that any covering problem is $(1,1)$-decomposable).

**Lemma 3.2.** *Set cover and Non-metric Facility Location are efficiently $(1, e/(e-1))$-decomposable.*

*Proof of Lemma 3.2.* Since NMFL is a generalisation of set cover, we only give the proof for NMFL. Through binary search, we may assume we know a budget $B$ such that the cheapest way to cover $i$ clients from the instance of NMFL has cost at most $B$. We implicitly convert our instance of NMFL into an instance of set cover (SC) with an *exponential* number of sets: for each facility $f$ and each subset of clients $f$ connects to, introduce a set of total cost equal to

the facility cost of $f$ plus the respective connection costs. Clearly, any partial solution to NMFL of cost at most $B$ covering at least $i$ clients corresponds to a partial solution to SC of equal cost and covering the same number of elements, and vice versa.

Now, we can convert this set cover instance into an instance of so-called *Budgeted Maximum Coverage Problem (BMC)*: In the BMC problem, given a collection $\mathcal{S}$ of sets defined over a universe of weighted elements, where each set $S \in \mathcal{S}$ has an associated cost $c(S)$, and given a budget $C$, the objective is to select a subcollection $\mathcal{S}' \subseteq \mathcal{S}$ such that the total cost satisfies $\sum_{S \in \mathcal{S}'} c(S) \leq C$, and the total weight of the elements covered by $\mathcal{S}'$ is maximized. Khuller et al. (30) presented a $(1 - 1/e)$-approximation algorithm for BMC.

Our set cover instance SC naturally converts into a BMC instance with budget $B$, and we can use the algorithm of (30). Crucially, their algorithm is greedy: in every stage, it considers the most cost-effective set, i.e. the set that covers the most uncovered elements per unit cost. In terms of the original instance of NMFL, this set corresponds to the following. Let $\mathsf{Unc} \subseteq C$ be the uncovered clients (elements), $D(f) \subseteq 2^{N(f)}$ a subset of the clients facility $f$ connects to $(N(f) \subseteq C)$, and denote by

$$\max_{(f, D(f))} \frac{\mathsf{cost}(f) + \sum_{c \in \mathsf{Unc} \cap D(f)} \mathsf{cost}(c, f)}{|D(f) \cap \mathsf{Unc}|}$$

the *effectiveness* of facility $f$. The facility-clients pair $(f, D)$ maximizing the effectiveness correspond to the most cost-effective set in SC, and vice versa. To see that this can be found efficiently, for every facility $f$, we order the respective clients it connects to by increasing order of connection cost, $c_f^1, c_f^2, \ldots$. For a given $f$ and given the cardinality of an optimal $D(f)$, it is clearly sufficient to only consider the first $|D(f)|$ clients in increasing order of connection cost. Hence this set can be found by comparing the effectiveness of $|F| \cdot |\mathsf{Unc}| \leq |F| \cdot |C|$ facility-clients pairs. $\qquad\square$

Note that this is asymptotically optimal, a $(1, e/(e-1)-\epsilon))$-decomposition in polynomial time implies a polynomial-time $(1 - O(\epsilon)) \ln |X|$-approximation to set cover, contradicting $\mathsf{P} \neq \mathsf{NP}$, (18). Extending this, we show a corresponding lower bound for online set cover with predictions.

**Theorem 3.3.** *No polynomial-time learning-augmented online algorithm for set cover achieves an $O(\log \eta \log |\mathcal{S}|)$-competitive ratio (or $o(\log k)$).*

*Proof.* Consider an offline instance of set cover $(X, \mathcal{S})$. We use such a learning-augmented online algorithm to solve it (present the elements of $X$ in arbitrary order). This results in an approximation guarantee of $O(\log |\mathcal{S}|)$. Since set cover does not admit a $O(\log |\mathcal{S}|)$- or a $(1 - \epsilon) \cdot \ln |X|$-approximation in polynomial time (assuming $\mathsf{P} \neq \mathsf{NP}$), see (39; 18), the proof follows. $\qquad\square$

## 3.2. Weighted Path Augmentation and Parking Permit Problem

The main result of this section is the following.

**Theorem 3.4.** *There is a deterministic, $O(\log \eta)$- and a randomized, $O(\log \log \eta)$-competitive learning-augmented online algorithm for weighted path augmentation and parking permit.*

The laminar structure of WPAP allows us to exactly compute the decomposition.

**Lemma 3.5.** *WPAP is efficiently $(1,1)$-decomposable.*

*Proof.* Consider an instance of WPAP with elements $E$, where the elements are indexed from 1 to $n$. For every $1 \leq i \leq n$ and in $\mathsf{poly}(n)$-time, we show how to compute a subset $A \subseteq E$ of size at least $i$, and a set of links in $L \subseteq \mathcal{L}$ that covers $A$, such that $\mathsf{cost}(L)$ is minimized. We handle this by dynamic programming. We define $\mathsf{DP}[i, j]$ to be the minimum cost to cover $i$ items from $j$ to $n$, only allowing links that do not contain elements from 1 to $j - 1$. If this is infeasible, set the value to $+\infty$. Clearly, when $i = n - j + 1$, this can be solved exactly, since the constraint matrix is totally unimodular, since it has the consecutive ones property (Chapter 19 of (40)). To update the table for other values, we proceed as follows. Denote by $c_{j,k}$ the cheapest cost of covering all elements between $j$ and $k$ (including the endpoints) with sets not including any element after $k$ (if $k < j$, this is 0). Again, $c_{j,k}$ can be computed exactly by total unimodularity. We set

$$\mathsf{DP}[i+1, j] := \min_{j \leq k \leq j+i} c_{j,k-1} + \mathsf{DP}[i - (k - j), k + 1].$$

Correctness follows from the observation that if the $k^{th}$ element $(i \leq k \leq n)$ is the first element not to be selected for a potential solution in $\mathsf{DP}[i + 1, j]$, the cost of covering elements $i$ to $k - 1$ is exactly $c_{j,k-1}$, and, the cost of selecting the remaining $i - (k - j)$ elements equals $\mathsf{DP}[i - (k - j), k + 1]$, since any link starting earlier than $k + 1$ includes $k$ and can be disregarded. The running time of this algorithm is $\mathsf{poly}(n)$ as $i$, $j$ and $k$ are at most $n$. $\qquad\square$

The authors in (36; 38) give a deterministic $O(K)$-competitive algorithm for WPAP, where $K$ is the number of different cost values for the links. $K$ can be assumed to be of order $O(\log |E|)$. Furthermore, in (36), they provide a *fractional*, $O(\log K)$-competitive algorithm for parking permit. In this setting, one has to maintain a monotonically increasing, fractional solution covering all arriving elements, i.e. the sum of the fractional values assigned to links covering each arrived element needs to sum up to at least one. To convert such a fractional solution into an integral solution, they provide a randomized *online rounding scheme* which only loses a constant factor (with respect to

the fractional solution). That is, they obtain a randomized $O(\log K)$-competitive algorithm for parking permit.

To use these results, we need to replace the dependence on $|E|$ with a dependence on $k$. To do so, we rely on the following technical lemma, which holds *irrespective* of the knowledge of $k$.

**Lemma 3.6.** *Consider an instance $(E, \mathcal{L}, \mathsf{cost})$ of WPAP. Up to losing a constant factor, one can assume that the links in $\mathcal{L}$ are* laminar *(i.e., the elements covered by any two links are either disjoint or contained in one another). Furthermore, to get a $O(\log k)$ (resp. $O(\log \log k)$) competitive algorithm, one can assume that there are at most $O(\log k)$ (resp. $O(\log^2 k)$) different cost classes.*

*Proof of Lemma 3.6.* Note that a guess on OPT can be maintained online, see e.g. (4). We now show that we can restrict to $O(\log^2 k)$ different cost classes in the setting where we use a $O(\log \log k)$-competitive for an instance of WPAP, irrespective of the knowledge of $k$. To this end, we maintain a 2-approximation guess on $\log \log k$, that is

$$\mathsf{guess} \leq \log \log k < 2 \cdot \mathsf{guess},$$

where $k$ denotes the number of elements seen so far. Whenever this exceeds $2 \cdot \mathsf{guess}$, we double our guess, i.e. $\mathsf{guess} \leftarrow 2 \cdot \mathsf{guess}$, and we re-run the algorithm from scratch. In each such run with guess $\mathsf{guess}_i$, the cost of the online algorithm is at most $\mathsf{guess}_i \cdot \mathsf{OPT}$. Since the $\{\mathsf{guess}_i\}_{i \geq 1}$ are geometrically increasing, this is dominated by the last run (which uses the correct estimate for $\log \log k$), hence the total cost incurred is at most $O(\log \log k) \cdot \mathsf{OPT}$. It remains to show that we may restrict to at most $O(\log^2 k)$ classes of links. To this end, set

$$C_i = 2^{2^{2 \cdot \mathsf{guess}_i}},$$

for each run. We claim that we may restrict to links with cost between OPT and $\mathsf{OPT}/C_i$. Indeed, all links of cost larger than OPT can be disregarded, and all links of $\mathsf{OPT}/C_i$ can be greedily chosen whenever an uncovered element arrives, incurring total cost at most OPT. This is since $C_i$ is strictly larger than the total number of arriving elements in the $i^{\text{th}}$ run. Hence, by rounding the cost of a link to the next power of 4, we may assume we have at most $\log(C_i) \leq \log^2 k$ different cost classes of links. The setting where we use a $O(\log k)$-competitive algorithm can be proved in an analogous way, though the number of different cost classes can then be restricted to $O(\log k)$.

We now proceed to show laminarity. For each cost class, we may remove links that are contained in other links, and short-cut the remaining ones as to make them disjoint. This loses a factor of 2 in the final approximation. Finally, to make the instance laminar, we proceed in increasing order of cost classes, from left to right. For any cost class, we start with the leftmost link, and add to it the (at most) two

links from the lower cost class that intersect its left and right boundary. We then shorten the next link in that cost class to make it disjoint from the previous link and proceed with it. Note that the cost of these new links has increased by at most a factor of 2: the cost of any resulting link at layer $j$ is at most that of the original link at that layer, plus the cost of two original links at layers $j-1, j-2, \ldots, 0$. Since the cost classes are geometrically decreasing, the claim follows. Finally, it remains to show that the resulting set of links is laminar. That is, for any two links $\ell_1, \ell_2$, if $\ell_1 \cap \ell_2 \neq \emptyset$, then either $\ell_1 \subseteq \ell_2$, or $\ell_2 \subseteq \ell_1$. To show this, we proceed by induction. For the base case, i.e. the links of smallest cost class, this is immediate. Hence, assume all resulting links of cost class $j-1$ and lower are laminar. Once we start processing the leftmost link at layer $j$, we add to it the two resulting links crossing its left and right boundary from layer $j-1$. Since both of these links are laminar with respect to links of cost classes smaller than $j-1$, and their union is a consecutive interval, the resulting link contains all links from lower levels that it intersects. Since we shorten the next link on layer $j$ to make it disjoint from this newly created link, the next link on layer $j$ does not intersect any links the previous link contains. All in all, this results in a laminar set of links, with cost that is at most $O(1)$ times higher. $\qquad\square$

**Lemma 3.7** (implicit in (36)). *Let $(E, \mathcal{L}, \mathsf{cost})$ be an instance with a laminar set of links $\mathcal{L}$. Then, for any monotonically increasing fractional solution of cost $\mathsf{cost}_{\mathsf{frac}}$, there exists an online rounding scheme with expected cost at most $O(1) \cdot \mathsf{cost}_{\mathsf{frac}}$.*

*Proof of Lemma 3.7.* Up to rounding to the next power of 2, we may assume that each link has weight being a power of 2. This loses a factor of 2 in the final approximation guarantee. For all such resulting weight classes of links, we remove the links which are entirely contained in another link. It follows then that each element $e$ is covered by at most one link in each layer. Denote by $x$ the fractional, monotonically increasing solution to the instance of WPAP. Since the elements arrive one-by-one and each element $e$ is covered by at most one link in each layer in $L := \{0, 1, \ldots\}$ (each of cost $2^i$), we index this monotonically increasing solution by $x_i(e)$, where $i \in L$. For the rounding scheme, we proceed as follows. Before the arrival of the first element, we draw a number $\tau \sim [0, 1]$ uniformly at random. Whenever element $e$ arrives, we update the solution $x$, and we find $i \in L$ such that

$$\sum_{j \geq i+1} x_j(e) < \tau, \text{ and, } \sum_{j \geq i} x_j(e) \geq \tau.$$

We then buy the unique link at level $i$ covering $e$. We claim that the expected cost (over the randomness of $\tau$) is

proportional to the cost of the fractional solution. To see this, consider some link $\ell$ at layer $i$. Denote by $e_{\text{first}}^{\ell}$ the first element to appear that lies inside $\ell$, and denote by $e_{\text{final}}^{\ell}$ the last element to appear that is covered by $\ell$. Note that this is well defined, since the adversary is oblivious, i.e. the input sequence is fixed. To buy link $\ell$ at some element $e \in \{e_{\text{first}}^{\ell}, \ldots, e_{\text{last}}^{\ell}\}$, it must hold that

$$\sum_{j \geq i+1} x_j(e_{\text{first}}^{\ell}) \overset{(*)}{\leq} \sum_{j \geq i+1} x_j(e) < \tau, \text{ and,}$$

$$\tau \leq \sum_{j \geq i} x_j(e) \overset{(*)}{\leq} \sum_{j \geq i} x_j(e_{\text{last}}^{\ell}).$$

Here, $(*)$ follows from laminarity and monotonicity. Hence, the probability of purchasing link $\ell$ at level $i$ is at most

$$\sum_{j \geq i} x_j(e_{\text{last}}^{\ell}) - \sum_{j \geq i+1} x_j(e_{\text{first}}^{\ell}).$$

Observing that the fractional value of a link only changes if the arriving element is contained in it, and denoting $\Delta x_j(e)$ as the *increase* in $x_j(\cdot)$ upon the arrival of element $e$, we can rewrite this as

$$x_i(e_{\text{last}}^{\ell}) + \sum_{j \geq i+1} \sum_{e \in \ell} \Delta x_j(e).$$

The total expected cost is thus upper bounded by

$$\sum_{i \in L} \sum_{\{\ell \mid \text{cost}(\ell) = 2^i\}} 2^i \left( x_i(e_{\text{last}}^{\ell}) + \sum_{j \geq i+1} \sum_{e \in \ell} \Delta x_j(e) \right).$$

Since each element is contained in at most one link per level, we can rewrite this as

$$\sum_{i \in L} \sum_{\{\ell \mid \text{cost}(\ell) = 2^i\}} 2^i x_i(e_{\text{last}}^{\ell}) +$$

$$\sum_{j \in L} \sum_{\{\ell \mid \text{cost}(\ell) = 2^j\}} \sum_{0 \leq i < j} 2^{j-i} \underbrace{\sum_{e \in \ell} \Delta x_j(e)}_{\leq x_j(e_{\text{last}}^{\ell})}$$

$$\leq O(1) \cdot \sum_{i \in L} \sum_{\{\ell \mid \text{cost}(\ell) = 2^i\}} 2^i \cdot x_i(e_{\text{last}}^{\ell}).$$

The right-hand-side is the cost of the fractional solution $x_i(\cdot)$, up to a constant factor. The lemma follows.

$\square$

**Theorem 3.8.** *There is a deterministic, $O(\log k)$-competitive algorithm for WPAP, as well as a randomized $O(\log \log k)$-competitive algorithm.*

*Proof of Theorem 3.8.* For the deterministic algorithm, by Lemma 3.6, we may assume that the instance of WPAP is

laminar and that there are at most $O(\log k)$ different cost classes. Since the deterministic algorithms of (36; 38) are $O(K)$-competitive with respect to the number of different cost classes, the claim follows. For the randomized algorithm, we proceed by randomized rounding. By the results in (15), it is possible to maintain a monotonically increasing fractional solution to an instance of set cover, that is $O(\log d)$-competitive, where $d$ is the maximum number of sets each element is contained in. In our setting, links correspond to sets and thus $d = O(\log^2 k)$. By Lemma 3.7, we can round this $O(\log \log k)$-competitive solution in an online fashion, incurring only constant factor loss. This yields the $O(\log \log k)$-competitive algorithm. $\square$

**Theorem 3.9.** *If $(E, \mathcal{L}, \text{cost})$ is not given in advance, there is a lower bound of $\Omega(k)$ on the competitive ratio of any deterministic or randomized online algorithm for WPAP.*

*Proof of Theorem 3.9.* The lower bound instance looks the same for both the deterministic and randomized setting: Each link has unit cost. The first link only covers the first element, the second link only covers the first and second element, the third covers the first, second and third element, and so on.

In the deterministic setting, the adversary releases the elements one-by-one, in order. Up to reordering the links, among all eligible links, we may assume that the online algorithm picks the one of smallest index. This yields the competitive ratio of $k = |E|$.

In the randomized setting, the adversary releases $k = \log |E|$ elements indexed by $\lfloor (1 - 2^{-j}) \cdot |E| \rfloor$, for $j \in \{1, 2, \ldots, \log |E|\}$, in that order. Note that whenever an uncovered element arrives, among all links covering it, the randomized online algorithm will select one such link uniformly at random. This is since all of the links remaining up to this point "look the same". We may also assume without loss of generality, that the online algorithm only buys a single link if an uncovered element appears. Hence, the probability that the $j^{\text{th}}$-element is not covered by a link covering elements $1, \ldots, j-1$ can be lower bounded by

$$(1 - 2^{-j}) \cdot (2^{-1} - 2^{-j}) \cdot \cdots \cdot (2^{-j+1} - 2^{-j}).$$

Rearranging and simplifying, we can further lower bound the probability of the online algorithm having to add a link to cover $e$ by

$$\prod_{i \geq 1} (1 - 2^{-i}).$$

Taking the logarithm, using that $\log(1 - x) \geq 2 \log(2) \cdot x$ and exponentiating, it follows that this is at least $1/4$. Hence, the lower bound of $\Omega(k)$ follows. $\square$

## 3.3. Metric facility Location and Steiner variants

The first result from this section is the following.

**Theorem 3.10.** *There is a polynomial-time, $O(\frac{\log \eta}{\log \log \eta})$-competitive learning-augmented algorithm for metric facility location.*

*Proof.* By the results of (35; 20), there is a deterministic $O(\frac{\log k}{\log \log k})$-competitive algorithm for facility location. Hence, this result follows in a black-box way from our framework, provided the problem is efficiently $(O(1), 1)$-decomposable. Decomposability follows from a problem that has been studied in the setting of *facility location with outliers*. In particular, in Theorem 5.2 of (17), the authors give a 3-approximation to the problem of serving a prescribed fraction of the clients as cheaply as possible, meaning that a $O(3, 1)$-decomposition can be computed in polynomial-time. $\square$

We now list other prominent problems that have received considerable attention in the network design community, where our framework applies. With the exception of (edge-weighted) Steiner tree, we do not know how to turn our approach polynomial, i.e. how to compute an $(O(1), 1)$-decomposition in polynomial time. Note that most of these problems generalize the set cover problem, hence this is impossible in polynomial-time.

**Theorem 3.11.** *Our framework provides a learning-augmented algorithm with competitive ratio $O(\rho(\eta, \cdot))$ for (i) Steiner tree and Steiner forest; (ii) Connected facility location; (iii) Group Steiner Tree and Forest; (iv) Node-weighted Steiner problems; (v) Directed Steiner problems.*

We instantiate our framework with the best known online algorithms for the corresponding underlying problems.

- Steiner tree and Steiner forest: These problems have $O(\log k)$-competitive online algorithms, (27; 12). These are the only two Steiner variants considered previously in the literature in our learning-augmented setting. Results were discussed in the introduction ((44; 9)).

- Connected facility location: This is a variation of the facility location problem, where the facilities need to remain connected. As such, it generalises the online Steiner tree problem. The best competitive ratio is $O(\log k)$, (42).

- Group Steiner Tree and Forest: The group setting is an important generalization of Steiner problems. Groups of terminals (terminal pairs) arrive one-by-one, and for each group, one needs to select one terminal (pair)

to be included in the Steiner Tree (forest). For edge-weighted graphs, the best competitive ratio is of order $O(\log^7 k \log^5 |V|)$, (37).

- Node-weighted Steiner problems: This generalizes (edge-weighted) Steiner problems and set cover problems. Problems are defined analogously, but the costs are on vertices (and edges). To use an edge, one needs to also have purchased its two incident vertices. The best achievable competitive ratio for Steiner problems is typically a poly-logarithmic factor higher than the edge-weighted setting, (37; 14), and for generalizations such as the group-setting, the best-known running time is *quasi-polynomial*, i.e. of the form $|V|^{\mathsf{poly}(\log |V|)}$.

- Directed Steiner problems: This further generalizes the node- and edge-weighted setting. In polynomial-time, there is a $O(k^\epsilon \log^{O(1)}(|V|))$-$(O(k^{1/2+\epsilon} \log^{O(1)}(|V|))$-) competitive algorithm for directed Steiner tree (forest), for any $\epsilon > 0$, (16).

In the remainder of the section, we show how to recover the results from (44) (and partially (9)).

**Theorem 3.12.** *Our framework provides a polynomial-time learning-augmented $O(\log \eta)$-competitive online algorithm for online Steiner tree.*

We make use of the $k$-MST and $k$-Steiner tree problem.

**Definition 3.13** ($k$-MST Problem). In the $k$-Minimum Spanning Tree problem, we are given an edge-weighted graph $G = (V, E)$, and a designated root vertex $r \in V$. The objective is to compute a tree rooted at $r$ that spans at least $k$ vertices while minimizing the total cost.

There exist several constant-factor approximation algorithms for the $k$-MST problem (see (7; 8; 13)), culminating in a 2-approximation algorithm presented in (22).

**Definition 3.14** ($k$-Steiner Tree Problem). In the $k$-Steiner Tree problem, we are given an edge-weighted graph $G = (V, E)$, where the vertex set is partitioned into terminals $\{r\} \cup T$ and Steiner nodes $S$, i.e., $V = T \cup S$. The objective is to compute a tree rooted at $r$ that spans at least $k$ terminals while minimizing the total cost of its edges.

**Lemma 3.15.** *There exists a 4-approximation for the minimum $k$-Steiner tree problem.*

*Proof.* We shortcut all Steiner nodes $S$, by passing to the *metric closure* restricted to terminals in $\{r\} \cup T$. That is, we consider the complete graph on vertex set $\{r\} \cup T$, where the cost of an edge between any pair of vertices corresponds to the cost of their shortest path in the original instance. Any tree can be mapped to the original instance with no greater cost. Conversely, by passing to the metric closure, the cost of any tree on terminals is increased by at most

a factor of 2, (31). Hence, using the 2-approximation for $k$-MST with root $r$ on the metric closure, and mapping back to the original graph, connects at least $k$ terminals with the root, and has total cost at most 4 times that of a minimum $k$-Steiner tree. □

*Proof of Theorem 3.12.* Lemma 3.15 shows that this problem is efficiently $(4, 1)$-decomposable. As there exists a $O(\log k)$-approximation for the Online Steiner Tree Problem (28), by Theorem 1.2 our framework provides a $O(\log \eta)$-competitive online algorithm for Steiner tree. □

## 4. Experimental evaluation

In this section, we evaluate the performance of our framework for the set cover problem. Our choice of set cover for experimental evaluation is motivated by practical considerations. The decomposition algorithms are efficient and admit simple implementations. Additionally, the existence of an accessible dataset makes Set Cover well suited for conducting scalable and meaningful experiments.

**Dataset** We use two datasets. The first one is the public dataset from the ongoing PACE Challenge, from the exact track on the Hitting Set Problem, (2) (this is equivalent to the Set Cover problem). On average, these instances have 2200 elements, and 1800 sets. The second dataset is randomly generated. Each instance consists of 1000 elements and 100 sets, each set contains 50 elements sampled uniformly from the groundset without replacement. Both datasets consist of 100 instances.

**Set-up and predictions** For every instance, we randomly fix 50% of its elements to be included in the prediction $\hat{X}$. We obtain the real instances $X$ from $\hat{X}$ by replacing an $\alpha$-fraction of the predicted elements by unpredicted elements, keeping the total number of arriving elements fixed. This results in the normalized prediction error $\eta/|\hat{X}| = 2 \cdot \alpha$. For each obtained instance $X$, we report the multiplicative gap between the solution of the greedy algorithm and that of an IP-solver (apx/opt).

**Evaluation** We use the standard $O(\log k \log |\mathcal{S}|)$-competitive algorithm from (4) to instantiate ICE (Algorithm 2), and also use it as our baseline (classical). We compare the baseline to the (averaged) performance of our approach: we either use an exact decomposition for $\hat{X}$ (ICE_exact), corresponding to Theorem 1.1, or an approximate decomposition (ICE_approx), corresponding to Theorem 1.2. Whenever the baseline algorithm is about to make an arbitrary choice among several tied sets, we use the decomposition to break the tie. This does not affect our theoretical guarantees in any way. We achieve substantial improvements over the baseline, even for large prediction errors. The effects between using an exact

or an approximate decomposition are negligible for the PACE dataset and quite small for the randomly generated instances, see Figure 2.

**Implementation** We use the Gurobi MIP-solver to compute the exact decomposition and the optima. They offer free academic licenses for researchers, (25). The rest is implemented in Python. The experiments were run on a High Performance Cluster, taking around 100 hours to complete.

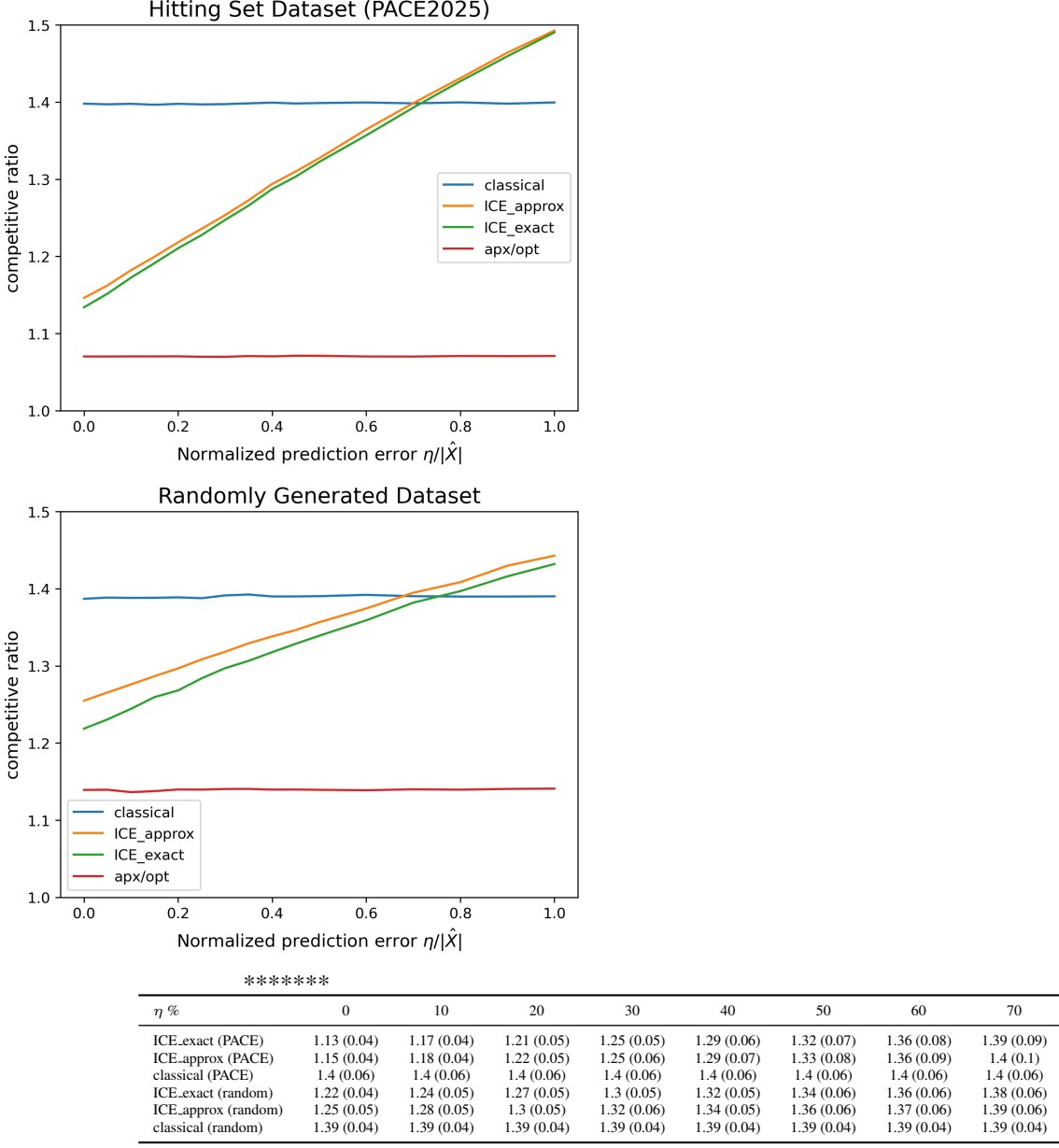

| $\eta$ % | 0 | 10 | 20 | 30 | 40 | 50 | 60 | 70 |
|---|---|---|---|---|---|---|---|---|
| ICE_exact (PACE) | 1.13 (0.04) | 1.17 (0.04) | 1.21 (0.05) | 1.25 (0.05) | 1.29 (0.06) | 1.32 (0.07) | 1.36 (0.08) | 1.39 (0.09) |
| ICE_approx (PACE) | 1.15 (0.04) | 1.18 (0.04) | 1.22 (0.05) | 1.25 (0.06) | 1.29 (0.07) | 1.33 (0.08) | 1.36 (0.09) | 1.4 (0.1) |
| classical (PACE) | 1.4 (0.06) | 1.4 (0.06) | 1.4 (0.06) | 1.4 (0.06) | 1.4 (0.06) | 1.4 (0.06) | 1.4 (0.06) | 1.4 (0.06) |
| ICE_exact (random) | 1.22 (0.04) | 1.24 (0.05) | 1.27 (0.05) | 1.3 (0.05) | 1.32 (0.05) | 1.34 (0.06) | 1.36 (0.06) | 1.38 (0.06) |
| ICE_approx (random) | 1.25 (0.05) | 1.28 (0.05) | 1.3 (0.05) | 1.32 (0.06) | 1.34 (0.05) | 1.36 (0.06) | 1.37 (0.06) | 1.39 (0.06) |
| classical (random) | 1.39 (0.04) | 1.39 (0.04) | 1.39 (0.04) | 1.39 (0.04) | 1.39 (0.04) | 1.39 (0.04) | 1.39 (0.04) | 1.39 (0.04) |

*Figure 2.* Competitive ratio for online set cover for varying prediction error $\eta$.

