# OpenReview forum: "Learning-Augmented Online Covering Problems"
_ICML.cc/2026/Conference — ICML 2026 regular_

### Official Review · Reviewer_FR6W · 2026-03-09

**Soundness:** 4
**Presentation:** 3
**Significance:** 4
**Originality:** 4
**Overall Recommendation:** 5
**Confidence:** 4

**Summary:**

The paper proposes a general framework for incorporating predictions into algorithms for online covering problems, with the goal of improving the competitive ratio based on the quality of the prediction.

In a general online covering problem, we are given a ground set and a family of subsets $\mathcal{S}$ together with a non-negative cost function on $2^{\mathcal{S}}$. Elements that must be covered arrive online one by one, and the algorithm must maintain a valid cover for all arrived elements at all times. Decisions about which sets to select are irrevocable. The objective is to minimize the total cost. Performance is measured by the competitive ratio, defined as the ratio between the cost of the online algorithm and the optimal offline cost.

Let $X$ denote the set of elements that arrive. The prediction model assumes that a prediction $\hat{X}$ of $X$ is given in advance. The prediction error is defined as $\eta = \min(|X|, |X \Delta \hat{X}|).$

The authors first show that if there exists an algorithm for an online covering problem with competitive ratio $\rho(|X|,\cdot)$, then, given a prediction $\hat{X}$, one can obtain a learning-augmented algorithm with competitive ratio $O(\rho(\eta,\cdot))$.

The approach is based on computing offline partial solutions that cover the predicted set $\hat{X}$. These partial coverings are incorporated into the online algorithm through a charging argument, where the cost incurred by the original algorithm is used to gradually purchase the partial solutions.

From a computational standpoint, the main bottleneck is computing this offline decomposition, which may not always be possible in polynomial time. To address this issue, the authors introduce the notion of a relaxed decomposition. They show that if such a relaxed decomposition can be computed in polynomial time for the offline version of the problem, then one can obtain a polynomial-time learning-augmented online algorithm. The competitive ratio in this case worsens by an additive term that depends on the quality of the relaxed decomposition compared to the $O(\rho(\eta,\cdot))$ guarantee.

The paper also includes experiments on both real and randomly generated datasets for the set cover problem.

**Compliance With Llm Reviewing Policy:**

Affirmed.

**Final Justification:**

My final recommendation is accept. The rebuttal did not change my evaluation, and the responses to other reviewers reinforced my beliefs about the paper.

**Key Questions For Authors:**

I do not have any questions for the authors.

**Limitations:**

yes

**Strengths And Weaknesses:**

**Soundness**

The paper appears technically sound. The prediction model and the error measure are reasonable, and the claims are supported by both rigorous theoretical arguments and experimental evaluation.

**Presentation**

The paper is well-written and easy to follow. The contributions are clearly stated, and the work is positioned appropriately within the existing literature.

**Significance**

Online covering problems form an important class of problems, and many online optimization problems can be expressed as a covering problem. The proposed approach is general and applicable to a broad range of such problems. A notable aspect of the framework is that the underlying online algorithm is used in a black box manner, meaning that its specific properties are not required as long as it fits the general structure of a covering problem. This makes the approach flexible and easy to apply. The framework itself is also simple and intuitive. In addition, the error metric is robust since it does not depend on the order in which elements arrive. Finally, the experimental evaluation includes results on real datasets.

**Originality**

One source of novelty of this framework is the prediction model, which assumes predictions about the set of arriving elements rather than the optimal solution. Additionally, for some covering problems this yields the first learning-augmented algorithm, while for others it leads to improved algorithms compared to existing learning-augmented approaches. This improvement stems from the fact that the underlying online algorithm is used in a black-box manner, without relying on specific structural properties of the algorithm being augmented. As a result, predictions can be incorporated into the best existing online algorithm for the problem.

---

> ### Author Rebuttal · Authors · 2026-03-30
>
> Dear Reviewer FR6W
>
> Thank you for your considerable time spent reviewing our paper and for your very positive comments.

---

> > ### Author Rebuttal · Reviewer_FR6W · 2026-04-02
> >
> > I do not have any concerns.

---

### Official Review · Reviewer_ettd · 2026-03-11

**Soundness:** 4
**Presentation:** 4
**Significance:** 2
**Originality:** 3
**Overall Recommendation:** 4
**Confidence:** 4

**Summary:**

"Algorithms powered by predictions" is a major impact area for ML. Here, (often online) algorithms get improved when we can make---with some possible errors---predictions about the paper based on the past, which then leads to better online algorithms. This paper explores in the context of a certain family of online covering problems, where the prediction is on the actual subset X of the domain that will need coverage over time.

**Compliance With Llm Reviewing Policy:**

Affirmed.

**Final Justification:**

The point about problems like DSF made me change my score.

**Key Questions For Authors:**

Can you extend your approach to problems where the dependence on k is much higher than (poly-)logarithmic, so that your improvements will be more significant?

**Limitations:**

Yes

**Strengths And Weaknesses:**

Soundness: The proofs and overall soundness look good to the best of my understanding. The experiments seem reasonable.

Presentation: The presentation is in general very good. I just have a few small comments:

(a) Use more standard notation for references, such as [34, 6].
(b) Intro: the function "cost" is defined on SUBSETS of the family of covers
(c) Make the definition of eta clearer using parenthesization as "(X \ X') union (X' \ X)"

Significance: This is where I have my biggest concerns. The dependence on k by the algorithms presented here is logarithmic, so unless eta scales as k^{o(1)} -- there is a "little-oh" in this exponent -- there is not that much improvement in the bounds. The error parameter eta being so small seems like a stretch to me, as a constant rate of relative error is perhaps more common. To me, this limits the paper's utility in its present form sharply.

Originality: The method is good, and is likely to be used/extended by others.

---

> ### Author Rebuttal · Authors · 2026-03-30
>
> Dear Reviewer ettd
>
> Thank you for your considerable time spent reviewing our paper and for your comments regarding the presentation. We will implement the small comments on the presentations that you are proposing.
>
> About your concerns on the significance of obtaining a dependence on $\eta$ (the prediction error) instead of $k$. There are indeed online covering problems, such as directed Steiner forest (DSF, see our full version) where the competitive ratio is of order $\tilde{O}(k^{1/2})$, and our LA-augmented algorithms yields a significant improvement, already for a constant rate of relative error (as you are proposing). We will actually stress this in the final version of the paper, thank you for pointing it out.
> Also, we want to note that for well-studied and broad problem classes, such as set cover, the logarithmic dependence is *tight* from a worst-case perspective and cannot be improved! That is why previous results in the literature also obtain results with logarithmic dependence. We emphasize that our paper is the first that (i) demonstrates *how* to learn the predictions - previous works use the black-box assumption that parts of an optimal solution can be learnt (and use heuristic approaches to find the prediction in the experiments section) and (ii) demonstrate how to leverage this information while compromising efficiency/competitiveness.
> Finally, our experiments empirically demonstrate that already with a constant fraction of correct predictions, we can obtain considerable improvements over the baseline algorithms. As you rightly point out, this is not explained by our theory (we would need a sublinear prediction error to bypass worst-case lower bounds). Still, this is interesting from a practical perspective and demonstrates that on average, our LA-framework lead to significantly better bounds than what can be explained by theory. This is also something we will highlight better in the final version.

---

> > ### Author Rebuttal · Reviewer_ettd · 2026-04-02
> >
> > The point about DSF is good. I still have concerns about eta, but will raise my score to "weak accept".

---

### Official Review · Reviewer_gYVB · 2026-03-12

**Soundness:** 3
**Presentation:** 3
**Significance:** 3
**Originality:** 3
**Overall Recommendation:** 4
**Confidence:** 3

**Summary:**

This paper studies online covering problems in a learning-augmented setting where the algorithm is given a prediction $\hat{X}$ of the set of requests that will arrive. The main contribution is a generic reduction, ICE (Iteratively Charge Expenses), that takes an online algorithm whose competitive ratio depends on the number of realized requests $k$, and converts it into a learning-augmented algorithm whose guarantee depends on the prediction error, defined by $\eta = min( |X| , |X Δ \hat{X}| )$

At a high level, ICE runs separate instances of the base algorithm on predicted vs. unpredicted requests and uses a simple thresholding (“rent-or-buy”) rule based on an offline decomposition of $\hat{X}$ into partial solutions $S_i$. The paper identifies conditions under which such decompositions exist (and can be approximated efficiently) and instantiates the framework for several classical covering problems, along with a small empirical evaluation for online set cover.

**Compliance With Llm Reviewing Policy:**

Affirmed.

**Final Justification:**

I still believe that the main issue with this otherwise very nice work is that it does not fully capture the notion of smoothness, as argued above. Overall, I believe my initial evaluation was fair, and I stand by it.

**Key Questions For Authors:**

**Questions for the authors**

- **Consistency–robustness optimality frontier:** A key motivation is a smooth consistency/robustness transition, but the paper does not seem to provide a general tradeoff impossibility/optimality statement of the form “for any γ-consistent algorithm, no σ-robustness better than … is achievable” (or an equivalent Pareto frontier). Are such frontier-style lower bounds meaningful in your covering-with-predicted-requests model? If yes, do you have any (even for special cases) that justify tightness of the η-dependence or the overhead terms? If no, could you clarify why this notion is not the right one here (e.g., because robustness is inherently pinned to the best-known ρ(k) baseline, or because η does not parameterize worst-case instances in a way that supports such a frontier)?

- **Error notion / smooth-tradeoff interpretation:** Your guarantees are stated in terms of η = min(|X|, |X Δ X̂|). Can you justify why symmetric-difference error is the right notion of “prediction quality” for covering, as opposed to a more solution-aware notion (e.g., overlap or cost-distance between an optimal solution for X̂ and one for X)? In particular, how should the reader interpret “smooth interpolation” in instances where |X Δ X̂| is large but the optimal structures for X and X̂ overlap heavily?

- **Asymmetry of prediction mistakes:** The analysis introduces separate quantities for false positives and false negatives (e.g., k⁻ = |X \ X̂| and k̄ = |X̂ \ X|). Can the final bounds be stated explicitly in terms of these asymmetric errors (or weighted errors), and if not, which step in the analysis forces the symmetric η summary?

- **Necessity of decomposition properties (A–D):** Which of properties (A)–(D) are essential for the analysis versus artifacts of the chosen charging/inductive proof? Can you provide a concise lemma/proposition showing that (α, φ)-decomposability (Def. 2.1) implies the existence of layers satisfying (A)–(D), with explicit constants, so the reader can see the logical dependency cleanly?


- **Scope of the framework assumptions:** The framework assumes union-closed/monotone feasibility and subadditive union costs. Where exactly are these used in the proof, and do you see meaningful extensions to non-monotone constraints (e.g., capacities/matroids/knapsack-style feasibility) or to cost models that are not subadditive under union?

- **Polytime vs existential applications:** For each listed application, can you explicitly state whether the result is (i) a fully polynomial-time learning-augmented online algorithm with a computable decomposition, or (ii) an existential guarantee assuming a decomposition oracle? A compact table listing (α, φ), decomposition runtime, and whether the result is implementable would help.


- **Experimental protocol clarity:** How is OPT computed in the experiments, and how are the online algorithms compared to OPT under a strictly online protocol? Also, can you report runtimes and add ablations isolating (i) decomposition quality (exact vs approximate), (ii) tie-breaking effects, and (iii) order effects or distributional shifts?

- **Baselines and comparisons:** Since APT is a key conceptual comparator, can you include an instantiated APT-style baseline where feasible (or explain why not), and/or compare to other recent predictions-on-requests approaches for relevant problems (e.g., online Steiner tree)?

**Limitations:**

yes

**Strengths And Weaknesses:**

Overall, I have some serious concerns about the modeling choices (notably the prediction error notion) and about verifiability/presentation in the current draft. That said, I find the proposed black-box ICE reduction and the (α, φ)-decomposition abstraction to be an interesting and potentially useful unifying direction for learning-augmented online covering, and the breadth of applications suggests the framework could be a valuable starting point for further work on black-box learning-augmented reductions. I therefore lean weak accept with moderate confidence (3/5), and my confidence could increase if the authors clarify the main conceptual questions (error notion, scope assumptions, and the Def. 2.1 → (A)–(D) connection) and in rebuttal.

**Strengths**

**Novelty and Innovation**
- **Black-box reduction:** Introduces a general, modular reduction (ICE) that converts competitive ratios depending on the number of realized requests k into guarantees depending on the request-prediction error η for a broad class of monotone covering problems.
- **Principled efficiency abstraction:** The (α, φ)-decomposition notion cleanly separates algorithmic efficiency from the achievable η-dependence, giving a structured explanation of why additive/overhead terms appear under computational constraints.
- **Simple, broadly applicable mechanism:** The two-instance ICE design (predicted vs unpredicted requests, with charging against offline layers) is conceptually simple yet widely reusable, and is positioned as avoiding subset-competitiveness or other algorithm-specific conditions used in some prior work.
- **Clean headline guarantee:** The results deliver an intuitive “replace k by η” message that aims to smoothly interpolate between accurate and inaccurate predictions while preserving worst-case robustness.

**Experimental rigor and Validation**
- **Initial empirical evidence:** Provides an experimental study for set cover (PACE + synthetic prediction noise), suggesting practical gains from predictions and showing small gaps between exact and approximate decompositions in the tested regimes.

**Clarity of presentation**
- **Clear motivation and setup:** The learning-augmented goal (consistency/robustness/smoothness) and the overall problem formulation are stated clearly, and the decomposition-and-charging intuition makes the high-level mechanism easy to understand.
- **Concrete application mapping:** The paper illustrates the framework on several canonical covering problems and states explicit parameter dependencies, which helps readers quickly interpret how η and problem size affect bounds.

**Significance of contributions**
- **Unifying view across core problems:** Unifies learning-augmented guarantees across foundational online covering problems (set cover, facility location variants, WPAP/parking permit, and several Steiner variants) under a single framework for predictions-on-requests.
- **Plug-and-play with best-known ratios:** Leverages best-known online algorithms in a black-box way to obtain improved η-dependent bounds (e.g., the stated O(log η / log log η) style dependence for metric facility location), potentially enabling quick adoption by theorists and practitioners.
- **Calibrated by barriers (set cover):** Includes a hardness discussion for set cover that helps set expectations about the limits of polynomial-time improvements.



**Weaknesses**

**Technical Concerns**

I begin with a **serious conceptual concern** that challenges the paper’s central “smooth tradeoff” motivation: the prediction error notion η = min(|X|, |X Δ X̂|) measures mismatch at the request-set level, but it may not reflect the actual value or harm of a prediction in covering problems. There are natural instances where the prediction is maximally wrong under symmetric difference (X ∩ X̂ = ∅), yet an optimal solution for the realized instance X also covers X̂ essentially for free, so trusting the prediction does not hurt and can still achieve the offline optimum. This makes the claimed “smooth transition” interpretation questionable unless further justified, or unless the guarantees are refined to depend on a more solution-aware error notion.

- **Error notion may be misaligned:** The capped, symmetric η can collapse very inaccurate predictions into the same regime and treats false positives and false negatives symmetrically even though their effect on covering cost can be highly asymmetric; it would strengthen the paper to justify η more carefully or provide variants parameterized by separate FP/FN measures or a solution-aware distance. (see Introduction / problem setup)
- **Lack of general optimality frontier:** The paper does not provide a general consistency–robustness Pareto frontier; the strongest “impossibility” evidence is mainly computational and application-specific (notably for set cover), which does not justify that the achieved η-dependence is broadly optimal across the framework. (see Section 3 hardness discussion)
- **Scope assumptions not clearly delineated:** The framework assumes union-closed/monotone feasibility and subadditive union costs; while natural for classic covering/network design, the paper does not clearly explain which important online settings are excluded (e.g., non-monotone constraints such as capacities/matroids/knapsack-like feasibility, or non-subadditive cost interactions) or whether relaxations are possible. (see Section 2 problem definition)
- **Decomposition requirements feel proof-engineered:** The analysis is phrased via decomposition properties (A)–(D), while polynomial-time implementability is stated via (α, φ)-decomposability; the connection is under-explained, and it is unclear which properties are essential vs artifacts of the charging/inductive proof. (see Def. 2.1 and Section 2.2)
- **Non-constructiveness and offline-oracle dependence:** For problems with exponentially large solution families, the existential decomposition is not constructive; the polytime story relies on nontrivial offline subroutines ((α, φ)-decomposability) whose availability, runtime, and overhead are not always made explicit. (see decomposition discussion + applications)
- **Proof-flow inconsistencies impede verification:** The analysis contains lemma/theorem numbering inconsistencies and self-referential citation patterns that make it harder to audit the proof flow. (see end of Section 2 analysis)

**Experimental Concerns**
- **Narrow empirical scope:** Experiments evaluate only set cover (PACE + synthetic prediction noise); other headline applications (facility location variants, WPAP, Steiner variants) lack empirical validation. (see Experiments section)
- **Ambiguous evaluation protocol:** The plots mix an “apx/opt” curve (e.g., greedy vs IP optimum) with online algorithm performance; it is not fully clear how the online algorithms and ICE variants are measured relative to OPT in a strictly online sense and under comparable computational budgets. (see Experiments section)
- **Missing robustness ablations:** There are limited ablations on prediction calibration, distributional shift, or order effects, and limited reporting on runtime and sensitivity to decomposition quality, even though decomposition is central to the approach. (see Experiments section)

**Presentation Issues**
- **Self-referential citation/numbering mismatch:** The proof block for Lemma 2.5 contains a self-referential citation (“From Lemma 2.5 and Claim 2.1 …”), and nearby, the text states “Theorem 2.6” while immediately following with “Proof of Theorem 2.7,” which suggests editing/numbering errors and complicates verification. (page 6, lines ~120–137; also echoed later in the same proof chain)

**Comparison and Related Work Issues**
- **Limited baseline comparisons:** While the comparison to Azar–Panigrahi–Touitou (APT) is conceptually useful, the empirical section lacks comparison to an instantiated APT-style baseline where feasible, and more generally lacks comparisons to recent “prediction-on-requests” approaches for relevant problems (e.g., online Steiner tree), which would strengthen positioning and the practical narrative.

---

> ### Author Rebuttal · Authors · 2026-03-31
>
> Dear Reviewer gYVB,
>
> Thank you for your comments and considerable time spent on our paper.
> We appreciate you pointing out the simplicity and plug&play mechanism with the currently best online algorithms for these problems - this is a key feature of our work.
>
> Error notion: Our error measure is simply the mismatch between the predicted and actually arriving elements (capped). While more tailored prediction models may exist for specific problems, we aim for a general framework applicable to a broad class of covering problems. From this perspective, our notion of $\eta$ very natural, and indeed has already been introduced and studied in prior literature.
> We deal with settings where guarantees depend on the number of requests. For example, our work replaces the $\log(k)$-term by $\log(\eta)$ in the competitive ratio of online Set Cover, where $\eta$ is never larger than $k$. As you point out, the covering cost can vary wildly among elements. However this does not affect worst-case guarantees, as they scale both offline and online solutions proportionally. To obtain a more solution-aware bound, already a finer approximation guarantee for the classical *offline* set cover problem would be necessary.
> For settings where requests differ but optimal solution overlap: in this case, instead of the requests, one should learn (parts of) the optimal solution, as in Bamas et al. Their prediction model improves on the $\log |\mathcal{S}|$-factor (but leaves the $\log k$-factor unimproved!). Still, as outlined at the end of page 3, we can combine their model with ours, obtaining the best of both worlds.
>
> Optimality frontier: our learning-augmented bounds simply match the currently best-known bounds for these classical online problems which are often already information-theoretically or computationally tight (for the whole range of $k$).
>
> Asymmetry of prediction mistakes: Indeed, throughout the analysis we treat false positives and false negatives separately. However, both types of errors ultimately lead to the same asymptotic dependence.
>
> Necessity of decomposition properties: it is always possible to decompose a covering problem according to (A)-(D), e.g. by simple enumeration. Whether or not this can be done in poly-time, is dictated by whether an $(\alpha, \gamma)$-decomposition can be computed in polynomial time. As such, it can't really be considered an artifact of the proof, but we agree it would be nice to give a simpler-to-describe decomposition - we did not managed.
>
> Scope of the framework assumptions: Non-monotone constraints such as capacities can't be handled since we heavily rely on the fact that we can stack partial solutions while remaining feasible. Subadditive union costs is also important as union of these partial solutions should not make the cost increase.  We will highlight this better in the final version.
>
> Polytime vs existential applications: We try to be as explicit as possible for which parameters of $\alpha$ and $\gamma$ a decomposition can be computed in poly-time. For problems with large/implicit solution spaces, we essentially require the same oracle access as the underlying offline problem (we did not have enough space to elaborate further). The distinction between polynomial and non-polynomial cases is stated in the lemmata rather than the theorems. The results on Set Cover and NMFL are polynomial time (it only requires exponential-time if we want to remove the additive dependence on $\log k$), for Section 3.2 it is always polynomial time, and for Section 3.3 we obtain a polynomial-time LA-algorithm for Steiner tree and metric facility location and exponential-time for the rest.
>
> Experimental Protocol Clarity: Our main takeaway is the improvement of our framework (with both exact and approximate decompositions) over the classical online algorithm. We focus on the exact $(1,1)$-decomposition and the efficiently computable approximate $(1,1-1/e)$-decomposition, as the latter is the best achievable in near-linear time.
> We report the 'apx/opt' ratio to control for instance difficulty: as we increase $\eta$ by modifying PACE2025 instances, we verify that the integrality gap (serving as a proxy for the instance hardness) remains essentially constant. This shows that improvements are not due to increasingly favorable instances.
> Finally, optimal solutions, as well as the exact decomposition for ICE_exact are computed with Gurobi, a state-of-the-art MIP-solver.
>
> Baselines and comparisons: We chose Set Cover for our experimental evaluation due to its practicality: the required decompositions are near-linear time, and suitable datasets are readily available. In contrast, implementing learning-augmented algorithms for more complex problems such as Steiner Tree is significantly more computationally demanding, despite being poly-time in theory (indeed, even prior work such as Azar et al. does not provide implementations).
>
> Finally, thank you for pointing out the numbering mismatch. We will fix that.

---

> > ### Author Rebuttal · Reviewer_gYVB · 2026-04-02
> >
> > Thank you to the authors for the thoughtful and helpful clarifications. The paper was genuinely **fun to read**, and I learned a lot from going through the proofs. I particularly appreciated the redefinition of the decomposition, and I view the work as a neat black-box reduction supported by strong mathematical analysis.
> >
> > The responses addressed my main concerns and clarified the technical points I was uncertain about. Overall, they made me more confident in my original assessment.
> >
> > My remaining hesitation is fairly minor and is mostly tied to how I interpret the prediction error notion and the resulting smoothness claim. More broadly, in learning-augmented works, I sometimes find it harder to assess whether the prediction error notion is an intrinsic one or one that is naturally shaped by the reformulation and the subsequent analysis. In this paper, the resulting notion also seems fairly connected to the matroid and submodular assumptions underlying the model. I do not mean this as a criticism of the present paper in particular, and I agree that the definition used here appears natural and well-motivated. My only reservation is that I am not yet fully convinced that this notion is as canonical as suggested in the main body.
> >
> > Relatedly, in robustness settings, I typically think of “smoothness” as reflecting a meaningful tradeoff in the worst-case guarantee as prediction quality degrades. I am not sure I completely see that interpretation in the same way here, perhaps because the setting is quite specialized and the prediction model itself is fairly discretized. It may be that a richer prediction object, for example a probability distribution over the likelihood of selecting each feasible set, possibly with some added randomization, would better capture a more gradual degradation, or at least lead to a cleaner tradeoff as the prediction error worsens.
> >
> > That said, the response addressed my concerns sufficiently, and I am now comfortable keeping my final score at **Weak Accept** now more confident. I have no further questions.

---

> > > ### Author Response · Authors · 2026-04-07
> > >
> > > Thank you again for the compliment, we are very happy to hear that the paper was well written and entertaining!
> > >
> > > The error notion (false positives plus false negatives) was already introduced in previous works. In particular it was shown, see e.g. Theorem D.1. of https://arxiv.org/pdf/2112.11831 (Online Graph Algorithms with Predictions), that the resulting error bounds (ours included) are tight for *any* variation of the false positives and false negatives making up $\eta$. In their paper, these correspond to $\Delta_1$ and $\Delta_2$, and their lower-bound construction applies to Steiner tree / forest problems, problems where our model applies as well. As such,
> > > we think that this error notion is the right metric to measure the competitive ratio of *worst-case* online algorithms with predictions. We will stress this also in the final version.
> > >
> > > About your second point on the "discretization" of prediction model. Our model can handle fractional predictions (e.g. probabilities corresponding to whether a request arrives or not), with a rather simple procedure: we obtain an integral prediction by simply "rounding" each request proportional to its probability. The resulting error $\eta$ (the error from the rounded prediction with respect to the actually arriving requests) is, in expectation, *exactly* the $\ell_1$-error between the predicted probabilities and the actual realization. The smaller (larger) the uncertainty, the smaller (larger) this expected error, i.e. the (expected) $\eta$. As such, the notion of smoothness is justified, and our model naturally handles very gradual changes in prediction quality. With small error $\eta$, our competitive ratios approach the offline optimum, and only slowly degrade as $\eta$ gets larger. On the other hand, we also have robustness: even if the predictions/probabilities are totally off, we are never worse than the worst-case competitive ratio of an online algorithm without predictions. For space reasons we have only sketched these arguments on l.210-218, but we agree that we should emphasize this more in the final version, other reviewers have also missed it. Thank you for replying to our rebuttal and raising this important point.

---

### Official Review · Reviewer_UjXW · 2026-03-13

**Soundness:** 3
**Presentation:** 2
**Significance:** 2
**Originality:** 2
**Overall Recommendation:** 3
**Confidence:** 3

**Summary:**

The paper studies learning-augmented algorithms for online covering problems (including set cover, facility location, Steiner tree variants, and path augmentation). The goal is to design algorithms that benefit from accurate predictions while maintaining worst-case guarantees when predictions are inaccurate.

The main contribution is a general black-box framework that converts any online covering algorithm into a learning-augmented algorithm whose competitive ratio depends on the number of requests $k$ and the prediction error $\eta$.

The framework relies on two main ingredients:

- an offline decomposition of the predicted request set, and
- an online procedure, called ICE (Iteratively Charge Expenses).

The authors show that this approach applies to several classical online covering problems and provide both theoretical guarantees and experimental results for the online set cover problem.

**Compliance With Llm Reviewing Policy:**

Affirmed.

**Key Questions For Authors:**

### 1. Computational complexity

Could the authors elaborate on the complexity of the decomposition step?

- What is the efficiency of computing $(\alpha, \gamma)$ decomposition for covering problems?
- What is the exact complexity of computing a $(1,1)$-decomposition?
- Are there references or known cases where this step can be implemented efficiently?

### 2. Prediction model

The framework assumes that the prediction is a set of requests. Could the framework be extended to settings where predictions provide:

- probabilities over requests, or
- partial information about future requests?

### 3. Tightness of the bounds

Are the competitive ratios achieved by the framework tight for the problems considered, or could improved bounds be obtained using problem-specific algorithms?

**Limitations:**

It is a theoretical work; no specific point needs to be discussed.

**Strengths And Weaknesses:**

## Strengths

### 1. General framework

The main strength of the paper is the generality of the proposed framework. The method is black-box, meaning it can transform a wide range of existing online covering algorithms into learning-augmented versions without modifying their internal algorithmic structure.

### 2. Explicit dependence on prediction error

The competitive ratio obtained by the framework depends explicitly on the prediction error $\eta$ and the number of requests $k$. This captures the standard desiderata for learning-augmented algorithms:

- consistency when predictions are accurate, and
- robustness when predictions are unreliable.

### 3. Broad applicability

The framework is applied to several classical problems:

- Set Cover
- Facility Location
- Parking Permit / Path Augmentation
- Steiner Tree variants

This breadth strengthens the claim that the paper provides a general design paradigm, rather than a problem-specific algorithm.

---

## Weaknesses

### 1. Computational complexity of the decomposition

A main limitation concerns the computational complexity of the decomposition step. In general, computing the required decomposition appears to be intractable. Even the $(1,1)$-decomposition requires a super-polynomial-time algorithm, as mentioned in the paper.

In the context of learning-augmented algorithms, predictions are typically assumed to be produced efficiently (e.g., via machine-learning models). Introducing an additional computationally expensive step may therefore undermine the practical benefits of using predictions.

### 2. Restrictive prediction model

The framework assumes that the prediction consists of the entire set of future requests. In practice, prediction models often provide:

- partial forecasts,
- probabilistic predictions,
- sequential predictions.

It is unclear how robust the framework is when applied to such more realistic prediction models.

---

## Additional Comments

1. When using the competitive ratio $\rho(\cdot,\cdot)$, it would be clearer to consistently use $k$ as the first parameter and $\eta$ as the second parameter (e.g., $\rho(k,\cdot)$ and $\rho(\cdot,\eta)$). Currently, both parameters sometimes appear in the first position.

2. Page 5 (end of the first column): it may improve clarity to use
   $\hat{X}_1, \hat{X}_2, \ldots$ instead of $X_1, X_2, \ldots$ for the predicted sets, for consistency with earlier notation.

---

> ### Author Rebuttal · Authors · 2026-03-30
>
> Dear Reviewer UjXW
>
> Thank you for your comments and considerable time spent on reviewing our paper.
>
> We would first like to address your first comments regarding the computational complexity including your additional questions regarding this issue.
>
> The complexity of computing an $(\alpha, \gamma)$-decomposition is problem specific and depends on the parameters $\alpha$ and $\gamma$. Lower $\alpha$ and $\gamma$ result in a better overall guarantees, but the decomposition may be harder to compute \textit{efficiently} ($(1,1)$ being the hardest one, but yielding the best guarantee). For problems like the parking permit problem, an $(1,1)$-decomposition can be computed in polynomial time (i.e. via a simple dynamic program as outlined in Lemma 3.5). For other problems like Set Cover and Facility Location, it is possible to compute a $(1, 1-1/e)$-decomposition in polynomial-time but no better (see e.g. the lower bound for polynomial time algorithms). Still, any problem admits an (1,1)-decomposition in exponential time, e.g. via enumeration.
>
> To summarize, let us stress the following point to avoid potential misunderstanding: a
> (1,1)-decomposition is the most refined (and hence, most difficult) decomposition to compute. For some problems (like the mentioned parking permit) this can be done efficiently, but often this requires super-polynomial running time. This is the case, e.g., for set cover, and note that this super-polynomiality cannot be avoided.
> Our introduction of $(\alpha, \gamma)$-decompositions is precisely meant to relax this requirement: this makes efficient computation possible for a broader class of problems, at the expense of slightly weakening the competitive ratio. For many problems (including set cover) this is indeed enough to get essentially the same competitive ratio but now with an efficient decomposition.
>
>
>
> With respect to the comment that predictions are typically assumed to be produced efficiently: also our predictions (set of arriving requests) can be learned very efficiently, and we give a rigorous method to leverage these predictions to improve existing guarantees. Our setting yields a clear improvement over several ML-augmented settings proposed in the literature, which often assume that part of an optimal solution can be learned from past instances. For computationally hard problems such as Facility Location or Set Cover, it is provably necessary to invest considerable computational effort (e.g., super-polynomial time) to obtain these optimal solutions. Many papers on ML-augmented online algorithms either consider problems where the underlying offline optimization is computationally easy (e.g., caching), or assume that these partially optimal solutions are given and focus on how to use them (which is still highly non-trivial). In this respect, we emphasize that our approach does not introduce an *additional* computationally expensive step, but rather, it is the other way around: in all settings where an approximate decomposition can be computed efficiently, (as for Set Cover), our framework allows us to remain efficient while exploiting prediction to improve the competitive ratio. We find that our prediction model bridges this gap cleanly and provides a more transparent account of the computational complexity.
> Moreover, in many cases, our model can be combined in a black-box manner with prior ML-augmented approaches, as outlined in Section 1.2 (e.g., for Set Cover and Facility Location).
>
> To your second concern about the prediction model. Our model can capture the first prediction model you allude to, e.g. the setting when probabilities over requests are provided. In this case, the metric on the prediction error is the expected error, i.e. the $\ell_1$ distance of the prediction to the actually arriving requests. Such a probabilistic prediction can simply be rounded while preserving the expected error, and then used with our model (and $\eta$ corresponds exactly to the expected prediction error). For the setting where partial information about future requests are gradually revealed, we do not know whether our model applies. This is a very nice model related to batch-arrival, though there is also only little work on this, even in the classical online setting.
>
> Finally, to your last question. The competitive ratios provided by our framework are optimal: since the prediction error is never larger than the number of arriving requests, these bounds exactly match the currently best algorithms for these online problems (which are often already information-theoretially optimal). In other words, any improvement would result in a better classical competitive online algorithm (e.g. with no prediction). This is a key feature of our work.

---

> > ### Author Rebuttal · Reviewer_UjXW · 2026-04-01
> >
> > I appreciate your responses and have no further questions at this time.

---

### Decision · Program_Chairs · 2026-04-30

**Decision:**

Accept (regular)

**Comment:**

This paper presents a general black-box framework for learning-augmented online covering problems. Reviewers have indicated the breadth of applicability, the simplicity and modularity of their approach, the explicit dependence of the guarantees on prediction error, and the ability to augment best-known online algorithms in a plug-and-play way across several classical problems.

The work was broadly seen as technically solid, well written, and potentially impactful as a unifying framework for future research in this area. Reviewers appreciated both the theoretical analysis and the empirical evaluations for set cover. Overall, despite some remaining questions about the prediction model and empirical evaluation, the paper received clearly positive support and merits acceptance.